# The concerted action of SEPT9 and EPLIN modulates the adhesion and migration of human fibroblasts

Matthias Hecht, Nane Alber, Pia Marhoffer, Nils Johnsson , Thomas Gronemeyer

Septins are cytoskeletal proteins that participate in cell adhesion, migration, and polarity establishment. The septin subunit SEPT9 directly interacts with the single LIM domain of epithelial protein lost in neoplasm (EPLIN), an actin-bundling protein. Using a human SEPT9 KO fibroblast cell line, we show that cell adhesion and migration are regulated by the interplay between both proteins. The low motility of SEPT9-depleted cells could be partly rescued by increased levels of EPLIN. The normal organization of actin-related filopodia and stress fibers was directly dependent on the expression level of SEPT9 and EPLIN. Increased levels of SEPT9 and EPLIN enhanced the size of focal adhesions in cell protrusions, correlating with stabilization of actin bundles. Conversely, decreased levels had the opposite effect. Our work thus establishes the interaction between SEPT9 and EPLIN as an important link between the septin and the actin cytoskeleton, influencing cell adhesion, motility, and migration.

## Introduction

Septins were discovered in the budding yeast in the 1970s and were subsequently found to form a novel cytoskeletal system (Longtine et al, 1996). Unlike actin filaments and microtubules, septins assemble into non-polar filaments. The mammalian genome encodes 13 different septins (SEPT1–SEPT12 and SEPT14) (Shuman & Momany, 2022). The basic septin building block in mammalian cells is a hetero-octamer composed of the SEPT2, SEPT6, SEPT7, and SEPT9 with a stoichiometry of 2:2:2:2 (2-7-6-9-9-6-7-2) (Mendonça et al, 2019). These building blocks can polymerize through end-to-end and lateral joining into higher ordered structures such as rings, filaments, and gauzes. Initially labeled as passive scaffold proteins, it has become more and more evident that septins are actively involved in many intracellular processes. They regulate vesicle transport and fusion, chromosome alignment and segregation, and cytokinesis (Surka et al, 2002; Bowen et al, 2011; Fuechtbauer et al, 2011; Estey et al, 2013; Tokhtaeva et al, 2015). In addition, septins cross-link and bend actin filaments into functional structures such

as contractile rings during cytokinesis, or stress fibers in filopodia and lamellipodia (Dolat et al, 2014; Mavrakis et al, 2014). Although actin and septin networks only partially overlap, they are structurally interdependent. Disruption of actin alters the organization of septin networks and vice versa.

Cell migration is mainly accomplished by a turnover of actomyosin stress fibers and focal adhesions. Septins are known to colocalize with and directly cross-link stress fibers and anchor these to the plasma membrane (Martins et al, 2023). They are enriched in the leading lamella and in radial actin stress fibers anchored to focal adhesions (Dolat et al, 2014). Several studies have linked septins to mechanotransduction, therefore playing a role in maintaining of front-rear polarity in migratory cells (Calvo et al, 2015; Simi et al, 2018; Lam & Calvo, 2019). Accordingly, the depletion of septins leads to a loss of the front-rear polarity axis in migrating cells (Shindo et al, 2018) and a SEPT9 knockout in mouse embryo fibroblasts impaired cell mobility (Fuechtbauer et al, 2011), whereas the overexpression of SEPT9 increased cell mobility in renal cells (Dolat et al, 2014). Loss of cell–cell adhesion and alteration of cell polarity are frequently observed in tumors of epithelial origin (Pal et al, 2018). These changes correlate with the altered SEPT9 expression levels in diverse types of epithelial cancer including prostate, breast, and colon cancer (Connolly et al, 2011; Gilad et al, 2015; Song & Li, 2015; Verdier-Pinard et al, 2017). Despite these numerous supporting observations, it is still elusive how SEPT9 is mechanistically linked to the adhesion and migration machinery.

The LIM domain–containing protein epithelial protein lost in neoplasm (EPLIN) is a modulator of cellular architecture and the actin cytoskeleton (Maul et al, 2003; Zhang et al, 2011) and was recently identified as a binding partner of SEPT9 (Hecht et al, 2019). EPLIN cross-links actin filaments into bundles and thereby inhibits the Arp2/3-mediated depolymerization and branching of filaments (Maul & Chang, 1999). Although the binding of EPLIN to the pointed end of actin filaments decreases depolymerization, the nucleation by Arp2/3 is inhibited, leading to a decrease in the dynamic turnover of the actin cytoskeleton. Low intracellular EPLIN levels correlate with an enhanced cancer cell invasion that is partially induced by a lack or loss of the tumor suppressor p53 (Ohashi et al, 2017). Epithelial–mesenchymal transition, characterized by a loss of

Institute of Molecular Genetics and Cell Biology, James Franck Ring N27, Ulm University, Ulm, Germany

Correspondence: thomas.gronemeyer@uni-ulm.de

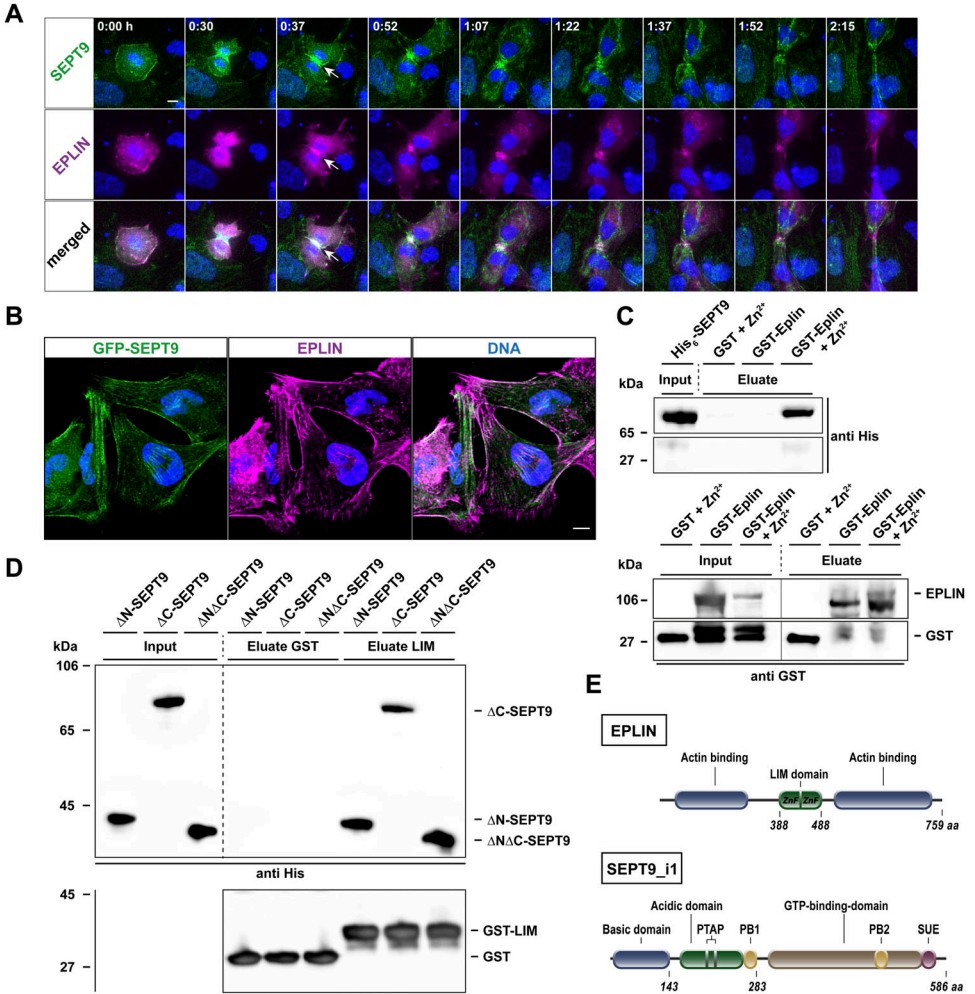

**Figure 1. SEPT9 colocalizes and interacts with epithelial protein lost in neoplasm (EPLIN).**
**(A)** GFP-SEPT9 and mRuby2-EPLIN colocalize at the cleavage furrow in dividing 1306 cells (white arrows). Confocal images were taken every 7.5 min and multiple Z-stacks combined to an average projection (scale bar = 10 $\mu$m). **(B)** Immunofluorescence of EPLIN in GFP-SEPT9 expressing 1306 fibroblasts showing colocalization along septin fibers and at the protrusion of cells (scale bar = 10 $\mu$m). **(C)** Pull-down showing the $Zn^{2+}$-dependent interaction of purified $His_6$-SEPT9 with GST alone and GST-EPLIN. A protein concentration of 1.5 $\mu$M SEPT9 and 0.1 mM $Zn^{2+}$ was applied. The line indicates two individual blots. **(D)** Purified, recombinantly expressed $His_6$-tagged SEPT9 fragments ($\Delta$N-SEPT9 [aa 295–586], $\Delta$C-SEPT9 [aa 1–567], $\Delta$N$\Delta$C-SEPT9 [aa 295–567]) were tested for binding to GST alone and GST-EPLIN$_{LIM}$ in a pull-down assay. The isolated G domain in the $\Delta$N$\Delta$C-SEPT9 construct was sufficient to mediate the interaction with the LIM domain. All assays were performed in the presence of 0.1 mM $Zn^{2+}$ with a protein concentration of 1.5 $\mu$M. The dashed line visually separates the input samples from the eluates. **(E)** Scheme of the domain structures of EPLIN and SEPT9i1 (PB, polybasic; SUE, septin unique element). Source data are available for this figure.

apico-basal polarity and abnormal alterations of cell shape and organization, is promoted by a reduction or loss of EPLIN (Zhang et al, 2011). The formation of adherens junctions requires a physical interaction of EPLIN with the cadherin–catenin complex (Abe & Takeichi, 2008). Furthermore, the overall maintenance of cell polarity in epithelial cells depends on the formation of a large protein complex comprising E-cadherin, $\beta$-catenin, $\alpha$-catenin, EPLIN, and F-actin (Chervin-Pétinot et al, 2012). EPLIN interacts with PINCH-1 at integrin adhesion sites leading to the recruitment of EPLIN to focal adhesions (Karaköse et al, 2015). EPLIN is also linked to the migratory machinery as its intracellular distribution is controlled by Rab40b/Cullin5 binding during cell migration. This protein complex regulates lamellipodium dynamics during cell migration (Linklater et al, 2021).

EPLIN is responsive to mechanical forces. However, because the LIM domain of EPLIN does not bind to actin, its mechanism of mechanosensing must differ from the LIM domain–dependent mechanisms of the paxillin and zyxin protein families (Taguchi et al, 2011; Gulino-Debrac, 2013).

We substantiate in this study the link between EPLIN and the septin cytoskeleton. We investigate the interaction between SEPT9

and EPLIN and aim to determine its role in cell adhesion and migration.

## Results

### SEPT9 interacts with the LIM domain of EPLIN

The EPLIN, the product of the LIMA1 gene, is a regulator of cell–cell/cell–surface adhesion and proliferation and was identified by us as a novel SEPT9 interaction partner in human skin fibroblasts (cell line 1306) (Hecht et al, 2019). To ensure consistency, we used here the same cell line. In addition, we validated selected results using the human foreskin fibroblast cell line BJ1-hTert (short BJH).

EPLIN and SEPT9 colocate not only along cytosolic septin fibers, but also at the protrusion tip of motile cells and at the cleavage furrow of dividing cells (Figs 1A and B and S1A, Video 1). The recombinant expression of both proteins in *E. coli* allowed us to show that purified SEPT9 interacts in a zinc-dependent manner with immobilized GST-EPLIN (Fig 1C). EPLIN is a largely unstructured protein with one central LIM domain (Fig 1E). The LIM domain

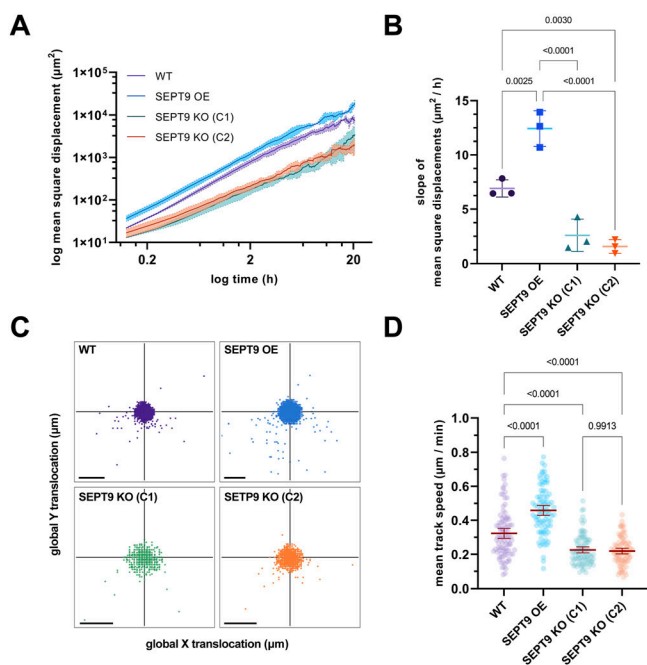

**Figure 2. SEPT9 overexpression increases cell motility.**
**(A)** Mean square displacement (MSD) of 1306 cells increases or decreases upon up- or down-regulation of SEPT9, respectively. Plotted are the log mean MSD ± range values from three independent experiments, each with n = 50 cells. **(B)** Slopes of MSD differ significantly between SEPT9 KO, WT, and SEPT9 OE cells. Significance values were calculated by one-way ANOVA followed by Tukey's multiple comparison test from three independent experiments, each with a sample size of n = 50 cells. Data of the three replicates are depicted as means ± SD. **(C)** Highest density of global step directions per cell line was the center, resulting in a random movement in the chemoattractant-free environment. Depicted is the individual translocation of each cell step in X and Y direction (scale bar = 50 $\mu$m). **(D)** Velocity of cell movement was significantly enhanced upon SEPT9 OE and significantly reduced upon SEPT9 KO. Significance values were calculated by one-way ANOVA followed by Dunnett's multiple comparison test from three independent datasets of total n = 89 cells. Depicted is the mean ± 95% confidence interval.

contains a double-zinc finger motif that is responsible for the correct folding of the domain (Michelsen et al, 1993). The zinc dependency of the interaction thus strongly supports that the LIM domain plays a role in binding to SEPT9. Furthermore, the interaction of EPLIN with SEPT2 was previously confirmed by Co-IP from HeLa cell lysates (Chircop et al, 2009). SEPT9 has a unique, long N-terminal extension preceding the conserved septin GTP-binding domain (short G domain) (Fig 1E). Given that SEPT2 and SEPT9 share only the G domain as a common feature, we investigated the direct binding of the SEPT9 G domain to the LIM domain of EPLIN. Indeed, the SEPT9$_{295-567}$ construct comprising solely the G domain is sufficient to bind the isolated LIM domain in a zinc-dependent manner (Fig 1D).

## The expression level of SEPT9 correlates with the migratory properties of fibroblasts

Increased levels of SEPT9 were shown to enhance the motility of murine cells and of human cancer cell lines (Füchtbauer et al, 2011; Marcus et al, 2019; Farrugia et al, 2020). We generated a 1306

fibroblast cell line stably overexpressing GFP-SEPT9 to study the effect of SEPT9 on cell motility. Stable overexpression resulted in an increase in SEPT9 levels by ~30% compared with WT cells (Fig S2A). To study the effect of the loss of SEPT9, we generated a CRISPR/ Cas9-mediated knockout (KO) of exons 4–6 in 1306 fibroblasts resulting in an interruption of the SEPT9 ORF (Fig S3A). The resulting cell line lacked detectable levels of SEPT9 as evaluated by Western blotting (Fig S3B–F). Cell motility of WT 1306 fibroblasts, cells overexpressing GFP-SEPT9 (SEPT9 OE), and two clones of our SEPT9 knockout (SEPT9 KO) cell line was compared by time-lapse microscopy combined with automated, artificial intelligence–assisted single-cell tracking. We used the log(MSD) (mean square displacement) over time as a measure of cell mobility. SEPT9 OE exhibited significantly higher log(MSD) values than the WT-level cells, whereas SEPT9 KO cells showed significantly lower log(MSD) values (Fig 2A and B). This indicates that SEPT9 expression levels in 1306 fibroblasts positively correlate with cell motility. Analysis of the global translocation directionality revealed random cell movement in the absence of chemoattractants for all investigated cell lines (Fig 2C). The velocity of SEPT9 OE was significantly enhanced, whereas the SEPT9 KO decreased the velocity of the cells below the WT level (Fig 2D). Because of the insignificant differences in cell mobility between different clones of the SEPT9 KO cell line, all subsequent experiments were performed only with clone C1.

To investigate the effects of the SEPT9 interactor EPLIN on cell migration, we constructed a cell line stably overexpressing GFP-EPLIN. Stable EPLIN overexpression led to an ~50% increase in intracellular EPLIN levels compared with WT cells (Fig S2B), whereas down-regulation through siRNA knockdown (KD cells) reduced the expression level below 10% in comparison with WT cells and cells transfected with a non-targeting siRNA (Fig S2C). We compared the overall morphology of these and our SEPT9 OE and KO cell lines with WT fibroblasts. SEPT9 OE exhibited an expanded cell morphology, whereas SEPT9 KO displayed a constricted morphology. EPLIN OE had no significant impact on the cell morphology, whereas EPLIN KD cells were rounded (Fig S4A and B).

The influence of SEPT9 and EPLIN on cell migration was investigated by a Boyden chamber assay. SEPT9 OE resulted in a 1.4-fold increase in the migratory potential of 1306 cells compared with WT cells (Fig 3, left). This effect was not observed in BJH cells because a large fraction of SEPT9-overexpressing cells were trapped in the Boyden chamber membrane pores, impeding quantitative analysis. The KO of SEPT9 induced an almost complete inhibition of cell migration (eightfold decrease) (Fig 3, left). The overexpression of EPLIN and EPLIN lacking the LIM domain did not significantly affect cell migration in 1306 and in BJH cells (Figs 3 and S5A). However, EPLIN KD cells showed a threefold higher migration rate as WT cells, or cells expressing a respective control siRNA (Fig 3, middle). To investigate the interdependency of SEPT9 and EPLIN interaction on cell migration, the Boyden chamber assay was performed with cells with both proteins simultaneously up- or down-regulated (Fig 3, right). The OE of EPLIN in SEPT9 KO cells resulted only in a minor restoration of the migration rate (4% migrated cells, compared with 1% in SEPT9 KO). In comparison with WT cells (9.7%), the already enhanced cell migration of SEPT9 OE (13.4%) could not be significantly increased by additional OE of EPLIN (14.3%). However, the elevated migration rate in EPLIN KD

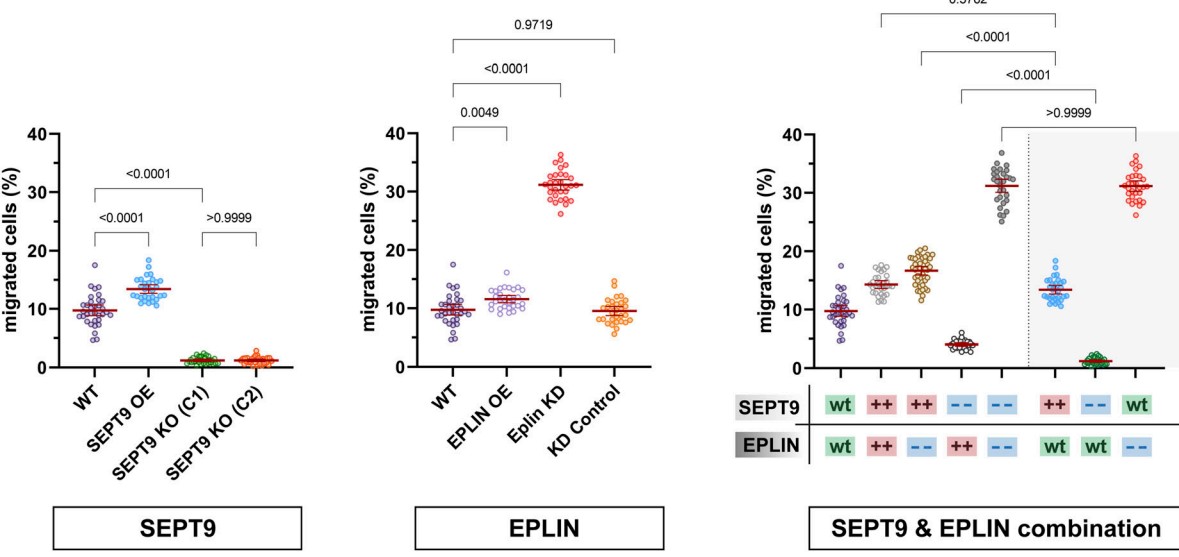

**Figure 3. SEPT9 and epithelial protein lost in neoplasm (EPLIN) coregulate cellular migration.**
SEPT9 OE migrate faster than WT cells, whereas SEPT9 KO cells do barely migrate at all (left). EPLIN OE does not affect migration, but EPLIN KD increases cell migration significantly (middle). EPLIN KD shifts the migratory potential of SEPT9 KO cells to the level of SEPT9 WT cells with EPLIN KD (right). All data points were gathered in a single experiment and separated into three protein level–dependent subpanels for improved data visibility. The gray highlighted data points (right panel) also show data from the left and center panels to allow appropriate comparability for combined protein-level variations within a single plot. Significance values were calculated by one-way ANOVA followed by Šidák's multiple comparison test from three independent experiments, each with n = 10 cells. All quantitative data are depicted as means ± 95% confidence interval.

cells (31.1%) was significantly reduced by simultaneous SEPT9 OE (16.6%) but remained unaffected by SEPT9 KO (31.2%). Taken together, the Boyden chamber assays demonstrated contrasting effects at low levels of SEPT9 (reduced motility) or EPLIN (enhanced motility). Conversely, high levels of SEPT9 could promote cell migration, whereas OE of EPLIN had no significant impact.

We next asked whether other cellular processes are also coregulated by SEPT9 and EPLIN. Migration involves breakdown and re-establishment of adhesion to the attachment matrix (Gardel et al, 2010). Malignant transformation is also correlated with the loss of cellular adhesion (Janiszewska et al, 2020). Considering that both proteins were associated with the metastasizing character of various cell lines (Zhang et al, 2011; Sun et al, 2015; Verdier-Pinard et al, 2017; Zeng et al, 2019; Farrugia et al, 2020), we monitored cell–surface adhesion under varying SEPT9 and EPLIN expression levels. Upon seeding of detached cells, the progress of reattachment and cell spreading was documented by light microscopy at regular intervals of 15–30 min. The fraction of attached cells and the degree of spreading were classified into five gradations ranging from 0% to 100%. First, the initiation of attachment for each cell line was determined (Fig 4A). When at least 25% of all cells attached to the surface, the elapsed time was considered as "initiation of attachment." EPLIN OE and EPLIN KD had no influence on the cell–surface interaction, as indicated by an initial attachment time of 12.5 min, which was almost identical to WT cells. In contrast, SEPT9 OE showed strong effects, reducing the time required for initial attachment to 7.5 min. SEPT9 KO cells exhibited a drastically prolonged attachment time of 52.5 min. Although the variation of EPLIN levels alone did not show any effects, the OE of EPLIN in SEPT9 KO cells could partially restore the WT-like attachment initiation to 20

min. A similar effect of EPLIN overexpression was observed for the adhesion progress of the SEPT9 KO cell line (Fig 4B and C).

The average time elapsed from seeding until substrate attachment of all cells was 105 min for WT, EPLIN OE, and EPLIN KD cells. Elevated levels of SEPT9 shortened this process to 60 min, whereas a SEPT9 KO severely delayed the process to 12–16 h. These effects were even enhanced when the progress of cell spreading was in addition taken into account (Fig 4D). To monitor spreading, the time was recorded until the normal morphology of at least 75% of the cell culture was restored. This spreading time was significantly reduced to 120 min in SEPT9 OE compared with 300 min in WT cells. In SEPT9 KO cells, the spreading process could not be properly measured because of the extensive time delay (18–22 h), but both reattachment and cell spreading were accelerated by EPLIN OE in these cells. Taken together, these results suggest an interplay between SEPT9 and EPLIN in the regulation of cell–surface interaction.

### SEPT9 influences the localization of EPLIN in cell protrusions

The interdependency of SEPT9 and EPLIN in cell adhesion and migration suggests an influence of SEPT9 on the subcellular localization of EPLIN or vice versa. Encouraged by a recent study highlighting the important role of EPLIN at the leading edge of migrating cells (Linklater et al, 2021), we focused on the subcellular localization of EPLIN and SEPT9 at cell protrusions of motile cells. Staining of endogenous EPLIN in stably GFP-SEPT9–expressing 1306 and BJH cells revealed a partial colocalization of EPLIN and SEPT9 at the tip of lamellipodia and along SEPT9 filaments (Figs 5A and S1C). In addition, EPLIN was observed in a dense belt-like subcortical network (Fig 5B and C). This EPLIN network around the cortex was

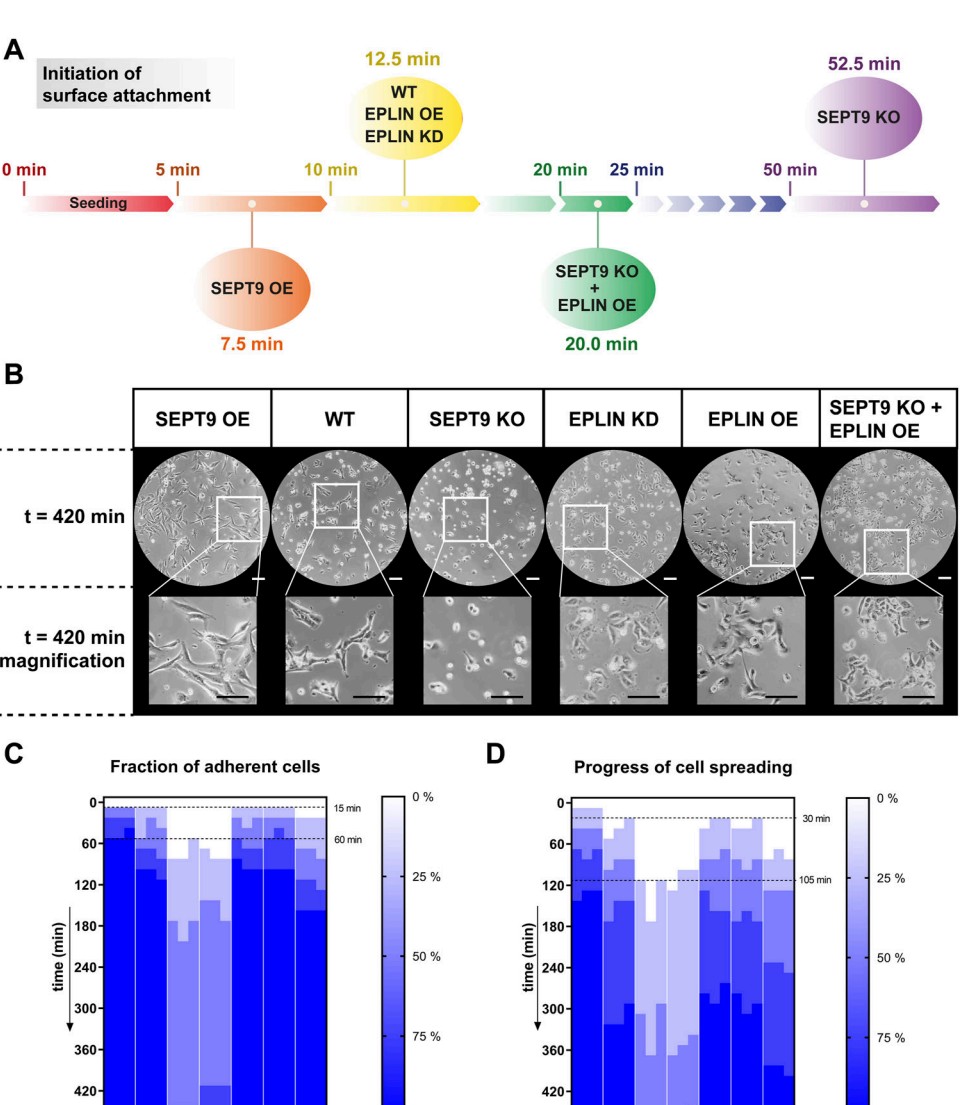

**Figure 4. Reattachment and spreading are tightly regulated by SEPT9 and epithelial protein lost in neoplasm (EPLIN).**
**(A)** Initiation of substrate reattachment was enhanced at high levels of SEPT9 and prolonged at low levels of SEPT9. Different expression levels of EPLIN did not influence this process. In the absence of SEPT9, an OE of EPLIN could partially rescue the delayed attachment of SEPT9 KO cells. **(B)** Reattachment assay revealed a positive correlation of SEPT9 concentration with cell adhesion and spreading. EPLIN alone did not influence the attachment and, however, could partially rescue the SEPT9 KO phenotype (10x objective; scale bar = 200 μm). **(C)** Heatmap representing the progress of cell–surface attachment. Based on light microscopy evaluation every 15–30 min, the reattachment was classified into five steps from 0% to 100%. **(D)** Heatmap of cell spreading upon seeding. The assays in (C, D) were performed in triplicate. At each timepoint, quantification was performed on 10 randomly selected positions with a shared phenotype for >80% of all cells. Cell adhesion and spreading were quantified by dividing the progress per replicate at each timepoint into five steps (0%, 25%, 50%, 75%, and 100%).

restricted by SEPT9 structures toward the cell center. Here, SEPT9 and EPLIN partially overlap by forming dense complexes or filamentous structures. We analyzed this subcortical EPLIN structure in SEPT9 KO cells and WT fibroblasts and quantified the relative change in its width. SEPT9 KO cells had a 50% reduced width of the EPLIN layer at the lamellipodium, whereas SEPT9 OE showed a 2.5-fold enhanced diameter (Fig 5D and E). EPLIN OE, however, induced the formation of eminently elongated filopodia along the whole-cell membrane (Fig 6A). These morphological changes varied from spike-like membrane protrusions to filament-like network formation within the cell. We quantified their length in cell lines displaying changing expression levels of SEPT9 and EPLIN (WT, EPLIN OE, EPLIN KD, SEPT9 KO, and SEPT9 OE) (Fig 6B and C; see Fig S2D and E for the relative expression levels). Filopodia were visualized by staining of the filopodium marker VASP and/or actin staining (Figs 6C and S5B, C, and F). The average length of filopodia increased by 1.7-fold upon EPLIN OE compared with WT and a control cell line overexpressing GFP. A similar result was observed in BJH cells (Fig S5B and C). The OE of SEPT9 did not significantly affect filopodium size. In contrast, the down-regulation of SEPT9 or EPLIN equally decreased the length of filopodia. The overexpression of EPLIN in SEPT9 KO cells restored the length of the filopodia to WT levels. To investigate whether the regulation of filopodium length depends on the interaction of both proteins, we measured filopodium length upon OE of a GFP-EPLIN construct lacking the SEPT9-binding LIM domain. EPLIN$_{\Delta LIM}$ did not induce elongated filopodia in 1306 or in BJH cells (Figs 6B–D and S5C, G, and H). In contrast, the mean size was reduced to levels observed in cells with a KD of EPLIN or a KO of SEPT9.

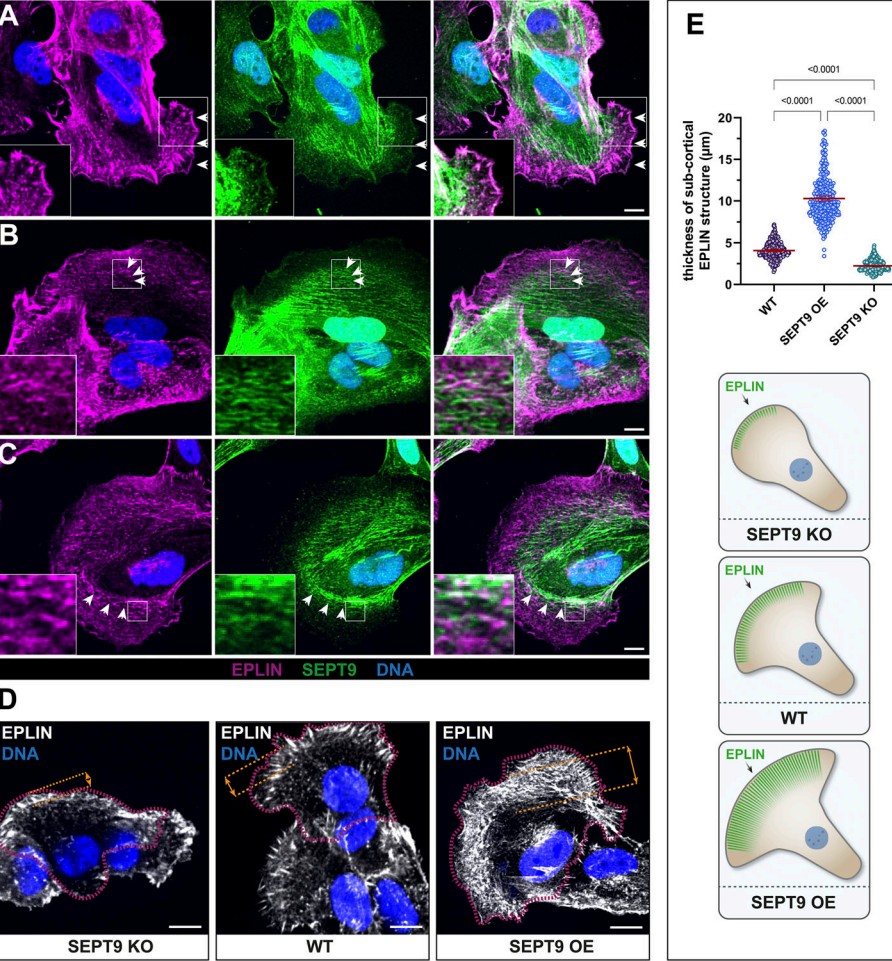

**Figure 5. SEPT9 and epithelial protein lost in neoplasm (EPLIN) colocalize directly or are adjacent in cell protrusions.**
Fluorescence microscopy images show the localization of immunostained EPLIN in 1306 cells expressing GFP-SEPT9. **(A)** Cell protrusion where SEPT9 and EPLIN colocalized at the tip of a lamellipodium. EPLIN is also localized along SEPT9 filaments in the cytoplasm. **(B, C)** Protrusions where the localization of EPLIN is restricted by the SEPT9 network. White arrows mark the location with a partial overlap of both proteins. **(D)** Cell protrusions showing the structural organization of EPLIN in relation to the SEPT9 expression level. Single cells are surrounded by red dotted lines. All images in (A, B, C, D) represent average intensity projections from confocal microscopy (scale bar = 10 $\mu$m). **(D, E)** Quantification of the EPLIN layer width in cell protrusions (as indicated in (D)) shows a significant increase upon SEPT9 OE and a significant decrease upon SEPT9 KO. These findings are illustrated in the cartoon. Significance values were calculated by one-way ANOVA followed by Tukey's multiple comparison test from three independent experiments, each with a sample size of n = 90 cells. The data are depicted as means ± 95% confidence interval.

The changes observed in filopodia as part of the migration machinery raised the question whether an impaired SEPT9-EPLIN interaction affects the overall migratory behavior of cells. Cell protrusions were tracked by actin staining combined with time-lapse microscopy. Protrusions in WT cells showed a directed distribution at the leading edge (Video 2). In contrast, cells with EPLIN KD cells showed multiple, branched cell protrusions assembling simultaneously in multiple directions (Video 3). To confirm again that an impaired SEPT9-EPLIN interaction is responsible for this cellular defect, GFP-EPLIN$_{\Delta LIM}$, lacking the SEPT9 interaction site, was overexpressed and monitored in migrating cells. Time-lapse microscopy revealed the simultaneous presence of cell protrusions, similar as observed in EPLIN KD cells (Video 4). Collectively, these experiments confirm the essential role of the interaction between EPLIN and SEPT9 in regulating the correct formation of protrusions at the cell tip during migration. Endogenous SEPT9 colocalized with filamentous GFP-EPLIN$_{\Delta LIM}$ and actin-containing structures at the cell cortex in a WT-like manner (Fig S5E and G).

## The interplay between SEPT9 and EPLIN regulates actin-based structures in fibroblasts

EPLIN was identified as an important member of the actin remodeling machinery (Collins et al, 2018; Taha et al, 2019). We next investigated the influence of SEPT9 and EPLIN on the actin cytoskeleton. In 1306 WT and BJH fibroblasts, the actin cytoskeleton consisted of transverse arcs or dorsal stress fibers at the cell cortex and ventral fibers toward the nucleus (Fig 7, center panel, and Fig S5B). Upon OE of SEPT9, the number and thickness of ventral stress fibers that span the whole cell increased considerably in both cell lines and SEPT9 colocalized with these fibers (Fig S1B), in line with published results (Dolat et al, 2014). These bundled fibers were associated with the focal adhesion marker paxillin (Fig 7, lower panel). In contrast, the KO of SEPT9 resulted in a complete loss of bundled actin filaments. The actin cytoskeleton in these cells consisted mainly of band-like structures along the plasma membrane (Fig 7, upper panel). EPLIN did not affect stress fiber formation but influenced the distribution of actin filaments at the cortex. The overexpression of EPLIN led to translocation of a portion

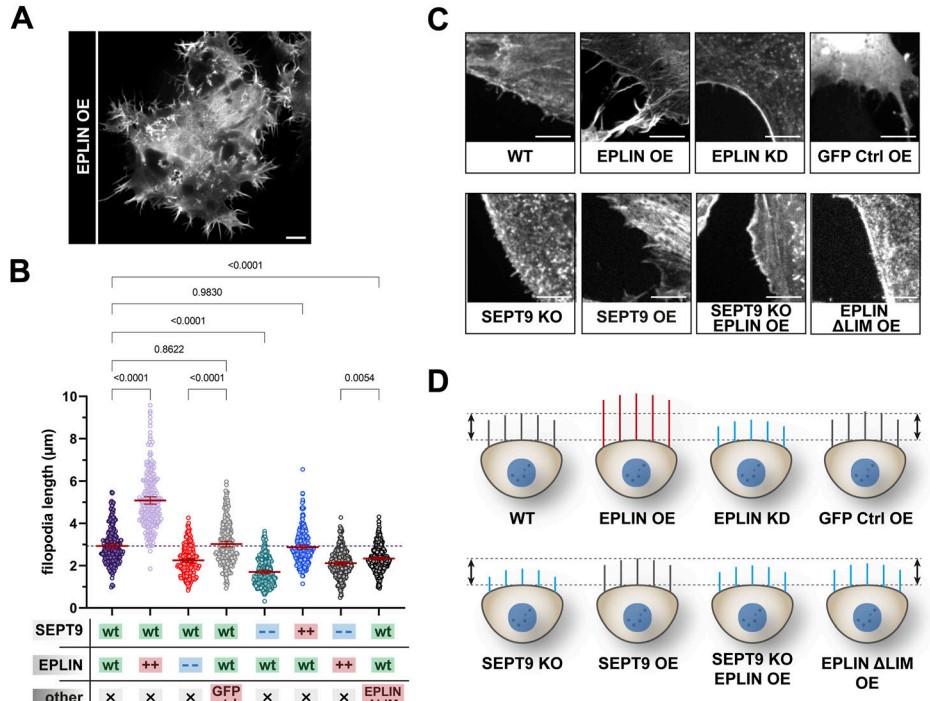

**Figure 6. Size of filopodia is dependent on epithelial protein lost in neoplasm (EPLIN) and SEPT9.**
**(A)** OE of GFP-EPLIN induced an enhanced formation of elongated filament-like or spike-like filopodia along the whole plasma membrane (scale bar = 10 μm). **(B)** Filopodium size of 1306 cells is dependent on EPLIN and SEPT9. An up-regulation of EPLIN significantly increased filopodium size, whereas the individual down-regulation of SEPT9 and EPLIN decreases their size. Neither the OE of EPLIN in SEPT9 KO cells nor the OE of an EPLIN mutant lacking the LIM domain showed elongated filopodia ("– –", knockout or knockdown; "++", overexpression). Analyses were performed 48 h upon the transient overexpression of the respective construct. Significance values were calculated by one-way ANOVA followed by Šidák's multiple comparison test from three independent experiments, each with a sample size of n = 80 cells. Data are depicted as means ± 95% confidence interval. **(C, D)** Confocal microscopy images showing representative sections of filopodia stained by phalloidin (scale bar = 50 μm) and (D) the corresponding schematic alteration. Filopodia in gray indicate a WT-like size, a decreased length is highlighted in blue, and increased filopodia are presented in red (relative filopodium lengths are drawn to scale).

of actin filaments toward the plasma membrane combined with the enhanced formation of filopodia. This effect was independent of SEPT9 (Fig 7, right column). A reduction of EPLIN (Fig 7, left column) led to a circular actin network with a fewer ventral stress fibers spanning the entire cell. Simultaneous SEPT9 OE maintained the circular organization but again enhanced the bundling of paxillin-associated actin filaments. The absence of both proteins abolished stress fibers and transverse arcs and left only thin filaments along the plasma membrane.

Taha et al suggested that EPLIN interacts with the Arp2/3 complex to regulate protrusion dynamics (Taha et al, 2019). Furthermore, an influence of Arp2/3 on cell adhesion and protrusion formation was reported (Beckham et al, 2014). However, we did not observe any noticeable effect of the Arp2/3 inhibitor CK666 on the EPLIN- or SEPT9-induced alterations of actin architecture or protrusion formation (Fig S6). The mobility of CK666-treated 1306 WT decreased by 76%, resulting in nearly immobile cells as detected by a Boyden chamber assay (Fig S7A). However, SEPT9 OE and EPLIN KD cells were less sensitive to CK666 treatment. Compared with DMSO-treated cells, the migration rate of both cell lines decreased by less than 10%.

EPLIN partially localizes along septin and actin filaments but is also found in the tip of lamellipodia and together with paxillin in focal adhesions. The down-regulation of EPLIN in 1306 fibroblasts not only reduced the size of focal adhesions but also caused the translocation of paxillin-rich structures from the membrane toward the cytoplasm (Fig 8A, left column). In contrast, the up-regulation of EPLIN strengthened paxillin structures at the plasma membrane (Fig 8A, right column). Both up- and down-regulation of SEPT9

resulted in preferential localization of paxillin at the plasma membrane with smaller adhesions upon SEPT9 KO and larger adhesions upon SEPT9 OE (Fig 8A, right column). Using paxillin as a marker, we subsequently quantified the size of focal adhesions upon up- or down-regulation of EPLIN and SEPT9 (Figs 8B–D and S1D). OE of SEPT9 increased the size of focal adhesions of 4.9 μm compared with the WT (3.7 μm). In contrast, the KO of SEPT9 led to a reduction in focal adhesion length (2.1 μm). Consistent with the increase and decrease in SEPT9, KD of EPLIN reduced the size of adhesions to 2.5 μm, whereas its OE elongated the size to 4.2 μm, though to a lesser extent than SEPT9.

To investigate the potential synergistic effects of EPLIN and SEPT9, we additionally measured the size change in a SEPT9 KO/ EPLIN KD, as well as in a SEPT9 OE/EPLIN OE cell line (Fig 8B). The double depletion reduced the size of focal adhesions further (1.6 μm), whereas the double OE had no severe impact (5.1 μm) compared with the individual protein OEs. Only the OE of SEPT9 in EPLIN KD cells (3.5 μm) was able to restore a focal adhesion length similar to the WT level. The OE of EPLIN in SEPT9 KO cells (2.2 μm) was not sufficient to induce larger adhesions. Their size remained at the same length as in SEPT9 KO cells with WT levels of EPLIN.

Migration defects were observed in cells with impaired SEPT9-EPLIN interaction including SEPT9 KO cells and cells expressing the EPLIN$_{\Delta LIM}$ construct. Compared with full-length EPLIN with increased paxillin structures, EPLIN$_{\Delta LIM}$ had an opposite effect by decreasing the size of focal adhesions (3.0 μm) (Fig 8F). Although this mutant of EPLIN could not reduce the focal adhesion size to the same extent as the complete KO of SEPT9 or KD of EPLIN, a significant decrease was observed compared with the WT (Fig 8E).

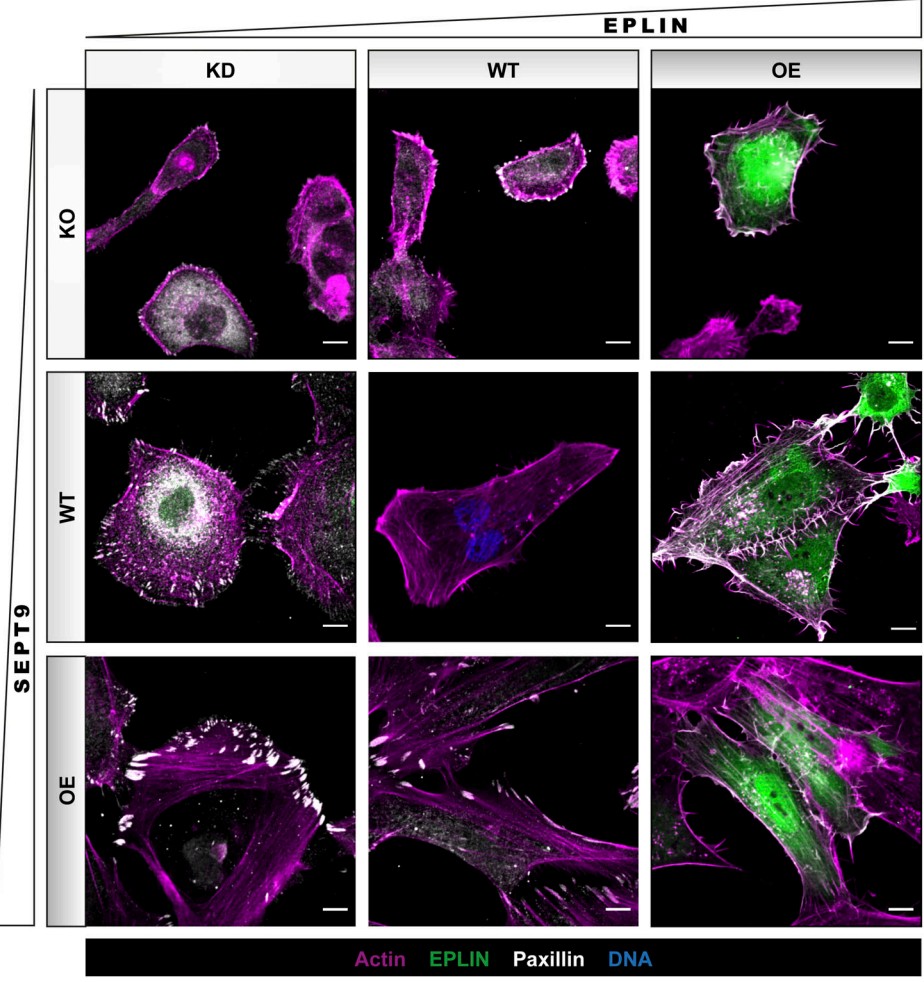

**Figure 7. Actin organization in 1306 fibroblasts is strongly dependent on SEPT9 and epithelial protein lost in neoplasm (EPLIN).**
In WT cells, actin (SiR-actin) is organized in filaments and bundled fibers throughout the cells and reaches into focal adhesions. The KO of SEPT9 destabilized actin filaments and induced an accumulation along the plasma membrane. An additional KD of EPLIN slightly enhanced this effect, whereas the OE of GFP-EPLIN showed no severe impact on the actin structure. Compared with WT cells, the depletion of EPLIN induced a circular organization of actin filaments and rounding of cells. The simultaneous OE of GFP-SEPT9 stabilized actin filaments and induced circular bundles. The OE of EPLIN alone favored the formation of long filopodia with actin filaments that were enriched along the membrane, also upon additional OE of SEPT9. The OE of SEPT9 alone induced long, straight stress fibers that reached into large focal adhesions. Maximum projections of confocal microscopy images are presented (scale bar = 10 $\mu$m).

Taken together, by identifying a correlation between length variation in paxillin structures and impaired migratory behavior of 1306 cells, we were able to show that the SEPT9-EPLIN interaction is essential in the regulation of migration via focal adhesions.

## Discussion

### SEPT9 and EPLIN act in concert to regulate cell migration and surface attachment

We previously identified EPLIN in a proteomics screen as a novel interaction partner of SEPT9 (Hecht et al, 2019). We confirm in this study that the G domain of SEPT9 is sufficient to mediate the zinc-dependent binding to the LIM domain of EPLIN in vitro. Furthermore, our findings demonstrate a strong dependence of EPLIN localization on SEPT9 throughout the entire cell cycle.

The positive correlation between cell movement and the expression level of SEPT9 suggested a crosstalk between the septin cytoskeleton and proteins that actively generate forces to induce changes in cell shape and motility. The nearly immobile SEPT9 KO cells emphasize the importance of a balanced SEPT9 regulation for

cell migration, consistent with previously published results in mouse fibroblast SEPT9 KO cells (Fuechtbauer et al, 2011) and in renal cells (Dolat et al, 2014). The concept of SEPT9-dependent mechanosensitive regulation is supported by the work of Yeh et al, who showed that SEPT9 is up-regulated in response to soft substrates, whereas stiff matrices induce a down-regulation of SEPT9 in endothelial cells (Yeh et al, 2012). Rather than manipulating the substrate stiffness and analyzing endogenous SEPT9 levels, we varied the level of SEPT9 and examined the adhesion and spreading of these cells. The time between cell seeding and the initiation of surface attachment was reduced (−40%) by high SEPT9 levels and severely prolonged (+320%) upon SEPT9 KO. These results were accompanied by a threefold faster attachment and spreading process in SEPT9 OE and a more than threefold slower rate in SEPT9 KO cells. We therefore propose a direct mechanistic link between the adhesive capability of a cell and the level of SEPT9, and suggest that EPLIN as an SEPT9 interaction partner represents this link. The elevated delay of cell–surface interaction in SEPT9 KO cells could partially be rescued by elevated levels of EPLIN. The deletion of EPLIN is associated with a destabilization of the cadherin complex (Abe & Takeichi, 2008; Taguchi et al, 2011). Elevated levels of EPLIN were shown to promote the formation of linear actin filaments by

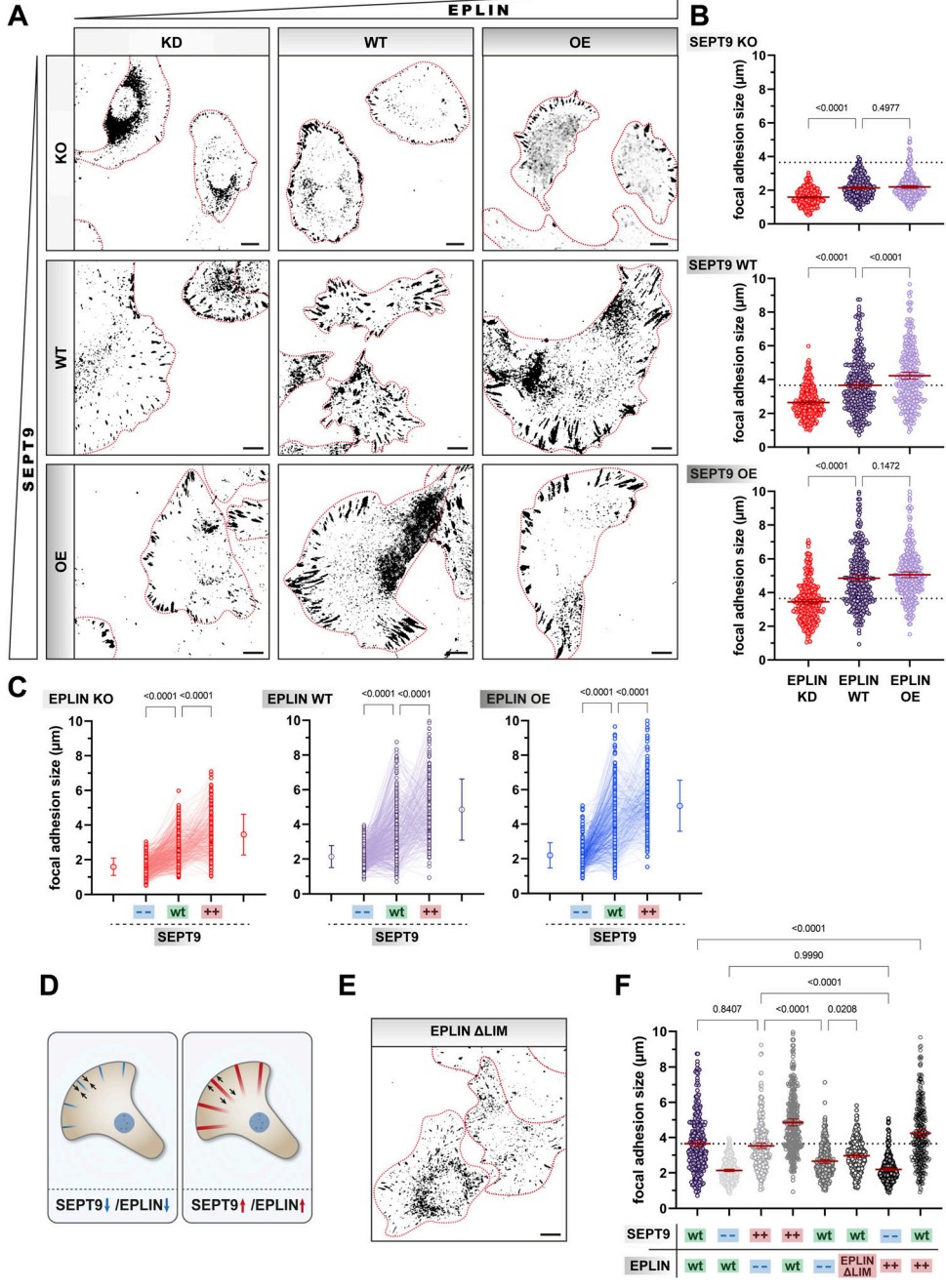

**Figure 8. Size and shape of focal adhesions are dependent on SEPT9 and epithelial protein lost in neoplasm (EPLIN).**
**(A)** Immunostaining of paxillin upon up- and down-regulation of EPLIN and SEPT9. For better visibility, paxillin is shown as an inverted image in black. Individual cells are outlined in red (scale bar = 10 $\mu$m). **(B, C)** Graphical representation of focal adhesion length in dependence of SEPT9 (B) and EPLIN (C) ("– –", knockout; "+ +", overexpression). Significance values were calculated by one-way ANOVA followed by Šidák's multiple comparison test from three independent experiments, each with a sample size of n = 100 cells. **(D)** Cartoon illustrating the reduction and increase in focal adhesions in dependence of SEPT9 or EPLIN. **(E)** Immunostaining of paxillin upon the overexpression of EPLIN$_{\Delta LIM}$. For better visibility, paxillin is shown as an inverted image in black. Individual cells are outlined in red (scale bar = 10 $\mu$m). **(F)** Quantification of the EPLIN$_{\Delta LIM}$ mutant revealed reduced focal adhesion size. In contrast, full-length EPLIN in WT cells increased the size of focal adhesions ("– –", knockdown; "+ +"/red field, overexpression). Significance values were calculated by one-way ANOVA followed by Dunnett's multiple comparison test from three independent experiments, each with a sample size of n = 100 cells. Quantitative data are depicted as means ± 95% confidence interval.

inhibiting the Arp2/3-mediated branching (Maul et al, 2003). This explains why the OE of EPLIN enlarges the cell surface area of SEPT9 KO cells. Providing a larger contact area between cell and substrate in combination with an enhanced actin filament formation rate reduces the time required for cell attachment and spreading. Our data suggest that initiation and progression of cell reattachment are not exclusively regulated by EPLIN because the OE or KD of EPLIN did not significantly alter the surface attachment compared with WT cells. Not only EPLIN is involved in cadherin-mediated cell adhesion (Chervin-Pétinot et al, 2012), but also other EPLIN-independent mechanisms such as integrin activation by talin are

crucial for cell adhesion (Lu et al, 2022). However, cellular migration was enhanced following EPLIN KD. EPLIN was previously reported as a cell migration inhibitor (Jiang et al, 2008; Liu et al, 2016; Collins et al, 2018), and accordingly, down-regulation of EPLIN is associated with higher migration rates.

Treatment of cells with the Arp2/3 inhibitor CK666 did not result in a significant alteration in actin filaments upon up- or down-regulation of EPLIN and SEPT9, respectively. In accordance with these observations, treatment with CK666 had only a minor effect on the motility of the cells. Our data thus contradict the model that EPLIN acts on actin filaments exclusively through the regulation of

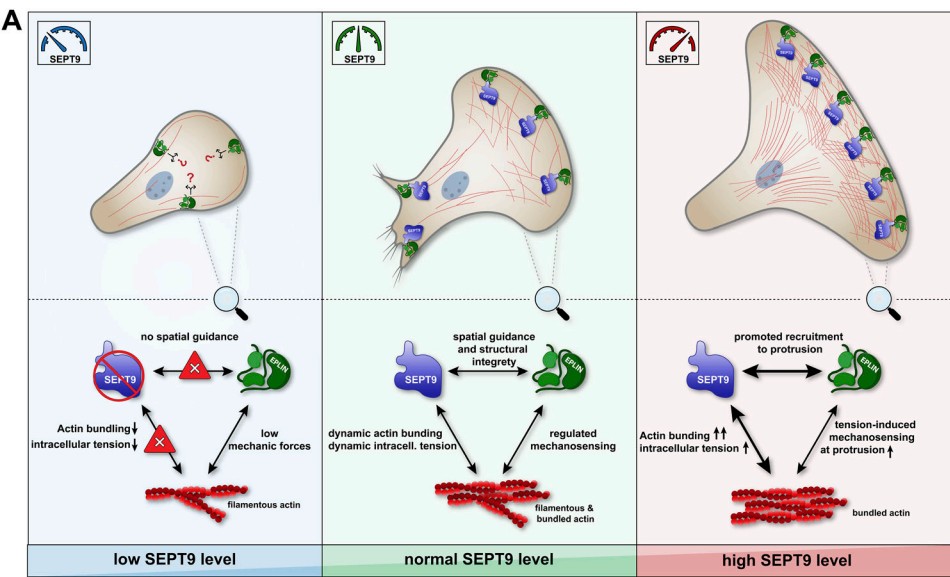

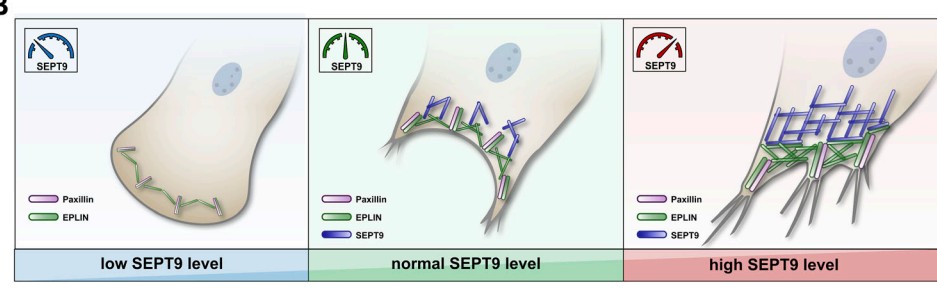

**Figure 9. Model of the mutual SEPT9 and epithelial protein lost in neoplasm (EPLIN) interplay in cell migration and focal adhesion regulation.**
**(A)** In WT cells, SEPT9 defines the mechanical properties of actin through its bundling activity. Regions of high tension (e.g., at the tip of migrating cells) are sensed by EPLIN to recruit the adhesion complex. SEPT9 may act as a scaffold for the localization of EPLIN along the lamella or recruits EPLIN-binding proteins to the front. Without SEPT9 (left panel), actin is less bundled and predominantly present in its filamentous state. Low-tension actin filaments cannot recruit EPLIN correctly to the leading edge. The enrichment of EPLIN at the cell tip is reduced, as absent SEPT9 cannot trigger its spatial recruitment. Elevated levels of SEPT9 (right panel) promote the bundling of actin (high-tension state) and the recruitment of EPLIN toward the cell front. Increased mechanical forces in the cell are sensed by EPLIN and "hyperactivate" the adhesion machinery. **(B)** Core multiprotein focal adhesion complex is assembled independently of SEPT9. The presence of SEPT9 is, however, responsible for the stabilization and growth of this complex mediated through actin and its direct interaction with EPLIN. The elevated level of SEPT9 (right panel) enhances the formation of octameric septin building blocks, which promote actin bundling and localize EPLIN to the sites of focal complexes. EPLIN presumably induces the recruitment of additional components such as paxillin, thereby promoting larger FAs. Reduced levels of SEPT9 (left panel) limit the amount of EPLIN at FAs and prevent the addition of further adhesion proteins. The complex is therefore limited to the minimal "core components," which allow the formation only of weak FAs.

Arp2/3. Instead, we suggest that the MAPK/ERK (mitogen-activated protein kinases/extracellular signal–regulated kinases) pathway is involved in EPLIN/SEPT9-regulated cell migration. Our MS screen identified two components of this pathway, MLCK and MYPT1, as interaction partners of SEPT9 (Hecht et al, 2019). Both proteins are involved in the formation and maintenance of lamellipodia and actin stress fibers (Joo & Yamada, 2014; Ghosh et al, 2021). This observation is supported by the finding that down-regulation of SEPT9 is associated with reduced activation of the MEK/ERK pathway in glioblastoma cells (Xu et al, 2018). Conversely, EPLIN is a direct downstream phosphorylation target of ERK (Han et al, 2007).

Increased levels of EPLIN induced the formation of spike-like and filamentous filopodia along the entire plasma membrane of the cell. A similar phenotype can be observed in fibroblasts expressing constitutively active Cdc42. Cdc42 activates the Arp2/3-mediated nucleation of branched actin, promoting polymerization in lamellipodia (Rohatgi et al, 2000). Mediated by formins, a synchronized elongation of parallel actin filaments induces the formation of filopodia. Similar phenotypes upon EPLIN and dominant-active Cdc42 OE point toward a common mechanism. EPLIN is required for the localization of Cdc42 and RhoA to the cleavage furrow during cytokinesis (Chircop et al, 2009), but further investigation is required to answer the question whether

EPLIN-mediated filopodium development is established by an enhanced interplay of these proteins or whether EPLIN indeed establishes a connection to the ERK signaling pathway, which is also known to directly regulate cell motility (Tanimura & Takeda, 2017).

We suggest that the regulation of cell adhesion and migration through EPLIN and SEPT9 is based on three key factors: (a) remodeling rate of the actin cytoskeleton, (b) bundling and stabilization of actin filaments, and (c) sensing of cellular tension. EPLIN was identified as a mechanosensor at adherens junctions (Taguchi et al, 2011). Our model proposes that the mechanosensitivity of EPLIN relies on the presence of SEPT9, which acts as a modulator of cellular tension and as a spatial guide for EPLIN.

The direct interaction between EPLIN and the cadherin–catenin complex connects intracellular forces to the adhesive machinery (Chervin-Pétinot et al, 2012).

Our model for the three scenarios SEPT9 KO, WT, and SEPT9 OE may explain how a balanced level of EPLIN regulates the remodeling of the actin cytoskeleton and thereby overall cellular dynamics (Fig 9). At WT SEPT9 levels, actin filaments are organized dynamically, providing stability and flexibility to the cell. In the presence of EPLIN, its mechanosensory function balances the adhesion machinery and the Arp2/3-mediated actin branching (Fig 9A, middle). Upon EPLIN KD, the cell loses its ability to sense extracellular forces or cytoskeletal tension, leading to enhanced actin

branching at the cell front and subsequent cell migration. The deletion of SEPT9 results in a reduction in bundled actin filaments and in stress fibers. EPLIN may sense the reduced cellular tension and limits the dynamics of the actin cytoskeleton (Fig 9A, left), resulting in a low motility rate (Fig 3C). Consequently, the KD of EPLIN and the resulting loss of its mechanosensing capability in SEPT9 KO cells induce enhanced motility (Fig 3C) even in the absence of stress fibers and actin filament bundles. SEPT9 OE shift the equilibrium of actin toward bundled filaments with high tension. The mechanosensing of a high-tension state through EPLIN promotes cell–surface adhesions to facilitate migration (Fig 9A, right). In SEPT9 OE, the KD of EPLIN does not induce an increase in cell motility (Fig 3C). The stabilization and mechanical support of actin bundles by SEPT9 limit actin branching, thereby inhibiting excessive cell migration as observed upon EPLIN KD in WT or SEPT9 KO cells.

### SEPT9 modulates the organization and localization of focal adhesions through EPLIN

At the lamella, where densely structured SEPT9 is separated from EPLIN, we observed a partial overlap along individual filaments. Cells with elevated levels of SEPT9 showed a denser network of EPLIN at the cell tip, whereas the absence of SEPT9 reduced the amount of EPLIN at the cell front, resulting in restricted cell migration. We propose that SEPT9 does not only act as a mediator of actin filament bundling but also promotes the accumulation of EPLIN at the leading edge. Colocalization studies in SEPT9 OE suggested that the SEPT9-rich lamella directly binds to EPLIN and regulates its local concentration. In epithelial cells, EPLIN responds to mechanical forces and is associated with the *zonula adherens* through its interaction with $\alpha$-catenin (Taguchi et al, 2011). Previous studies linked the intracellular expression level of SEPT9 to the morphology of focal adhesions (Fuechtbauer et al, 2011; Dolat et al, 2014). By showing an increase in focal adhesion number and length with increasing levels of SEPT9 and EPLIN, our data incorporate EPLIN in this pathway. In the absence of SEPT9 or EPLIN, focal adhesions were significantly reduced in size. The cumulative reduction in focal adhesion size was observed in double-depleted cells (EPLIN KD in SEPT9 KO cells). These findings suggest that the presence of both proteins is required for maintaining a stable focal adhesion complex.

Similar to cells with EPLIN KD, cells overexpressing EPLIN$_{\Delta LIM}$, lacking the SEPT9 interaction site, showed on the one hand a reduced focal adhesion size and on the other hand a re-localization from the membrane toward the cytoplasm. Taken these findings together, we propose that EPLIN binds to the focal adhesion machinery independently of SEPT9 through its known interaction partners $\alpha$-catenin (Chervin-Pétinot et al, 2012) and paxillin (Kasai et al, 2018). However, the intracellular amount of SEPT9 appears to influence the local concentration of EPLIN. As SEPT9 levels increase, the septin cytoskeleton at the lamella recruits actin filaments, thereby promoting the local concentration of EPLIN. Subsequently, this enhanced local enrichment of EPLIN at the leading edge would trigger a cascade to recruit additional binding partners of the adhesion complex (Fig 9B, right). Decreasing levels of SEPT9 restrict the formation of actin bundles, thereby limiting the localization of EPLIN to focal adhesions. Lowered concentrations of EPLIN cannot recruit

additional focal adhesion components, resulting in smaller and fewer adhesion complexes. Reduced adhesive structures result in a decreased contact area with the surface, thereby weakening the adhesive and migratory behavior of SEPT9 KO cells (Fig 9B, left).

The collective interplay of over 100 different components contributes to motility and adhesion. It is likely that the list of participating proteins is still incomplete, and their roles are not yet fully understood. Our data demonstrate the contribution of SEPT9, through its interaction with EPLIN and actin, to both processes.

## Materials and Methods

### Plasmids, cloning, and siRNA

All used and generated plasmids are summarized in Table 1. Cloning was performed using standard procedures. The ORF of EPLIN isoform alpha was PCR-amplified from Addgene plasmid #40928 to generate plasmid #1. All other EPLIN constructs were derived from this plasmid. All SEPT9 constructs were derived from plasmid #5 (a kind gift from *E. Spiliotis*, University of Virginia, Charlottesville, VA, USA). The iRFP ORF was PCR-amplified from Addgene plasmid #45457. All used PCR primers are listed in Table S1. siRNA targeting the LIM domain of EPLIN (# SASI_Hs02_00326071; sequence start 1,320; TATTGTAAGCCTCACTTCAA) and a scrambled control siRNA (#SIC001) were obtained from Sigma-Aldrich. The specificity of the employed siRNA was assessed by a second siRNA in a transwell migration assay (see below) (# SASI_Hs02_00326076; sequence start 1822; CCATTCACTGTAGCAGCTT). 1306 WT cells treated with this siRNA migrated 1.6 times faster than cells treated with scrambled siRNA (Fig S7B).

### Expression of recombinant proteins and in vitro pull-down experiments

All recombinant proteins were expressed in *E. coli* BL21DE3 in super broth (SB) medium. Protein expression was induced with 0.1 mM IPTG (supplemented with 2% vol/vol ethanol for SEPT9 expression constructs) at an O.D.$_{600}$ of 0.8, and protein expression was conducted for 5 h (EPLIN constructs) or overnight (SEPT9 constructs) at 18°C. The cells were harvested by centrifugation and stored at –80°C until further use. Cell lysis was performed in appropriate buffer (see below) by treatment with 1 mg/ml lysozyme and ultrasound, and cell debris was removed by centrifugation (40,000$g$, 10 min). For His$_6$-tagged proteins, IMAC buffer A (50 mM KH$_2$PO$_4$, 20 mM imidazole, 300 mM NaCl, pH 7.5) supplemented with e-complete protease inhibitor cocktail (Roche) was used as lysis buffer. These proteins were purified using a 5-ml HisTrap Excel column (Cytiva) mounted on an Äkta Pure chromatography device (Cytiva). Elution was carried out with a step gradient consisting of consecutive steps of 15%, 30%, and 100% IMAC elution buffer (50 mM KH$_2$PO$_4$, 200 mM imidazole, 300 mM NaCl, pH 7.5). The proteins were subsequently buffered in PBS using a PD10 desalting column (Cytiva), concentrated, and used for pull-down. GST-fusion proteins were directly lysed in PBS supplemented with e-complete protease inhibitor cocktail as described above. The resulting extract was

**Table 1. List of plasmids.**

| | Construct | Vector backbone | Insert | Source |
|---|---|---|---|---|
| #1 | EGFP-EPLIN | pLenti-CMV-GFP-Puro | fl (isoform alpha) | This study |
| #2 | EGFP-EPLIN LIM | pLenti-CMV-GFP-Puro | aa 228–288 | This study |
| #3 | EGFP-EPLIN (ΔLIM) | pLenti-CMV-GFP-Puro | Δaa 228–288 | This study |
| #4 | mRuby-EPLIN | mRuby2-C1 (#55911; Addgene) | fl | This study |
| #5 | EGFP-SEPT9 | pEGFP-C2 | fl | *E. Spiliotis* |
| #6 | GST-EPLIN | pGEX2T | fl | This study |
| #7 | GST-EPLIN LIM | pGEX2T | aa 228–288 | This study |
| #8 | GST-SEPT9 | pAc-GST | fl | This study |
| #9 | His-EPLIN | pES | fl | This study |
| #10 | His-SEPT9 | pES | fl | This study |
| #11 | His-SEPT9 ΔC | pES | aa 1–567 | This study |
| #12 | His-SEPT9 ΔN | pES | aa 295–586 | This study |
| #13 | His-SEPT9 ΔN ΔC (G domain) | pES | aa 295–567 | This study |
| #14 | pSpCas9(BB)-2A-iRFP670 | pSpCas9(BB)2A-GFP (#48138; Addgene) | iRFP670 | This study |
| #15 | pSpCas9(BB)-2A-GFP-Ex4B | pSpCas9(BB)2A-GFP (#48138; Addgene) | SEPT9 exon 4 sgRNA | This study |
| #16 | pSpCas9(BB)-2A-iRFP-Ex6E | #14 | SEPT9 exon 6 sgRNA | This study |

pAC-GST and pES are in-house–constructed pET15A-based plasmids for the expression of $His_6$-tagged proteins in *E. coli*.

immediately used for pull-down experiments. For each sample, 50 $\mu$l of PBS-equilibrated glutathione Sepharose beads (Cytiva) was incubated for 20 min with 500–1,000 $\mu$l extract of the respective GST-tagged protein (or 500 $\mu$l of 2 $\mu$M purified GST for controls) under rotating agitation. After washing with PBS, non-specific binding sites were blocked by incubation with 1.5 $\mu$M BSA for 15 min. Purified $His_6$-tagged proteins were added at a concentration of 1.5 $\mu$M and incubated for 30 min. Unbound proteins were washed away with PBS before an elution step with GST-elution buffer (50 mM Tris, pH 8.0, 10 mM reduced glutathione) for 10 min. Samples of the elutates and input proteins were subsequently analyzed with SDS–PAGE and Western blot, respectively. SDS–PAGE was performed using Bolt 4–12% Bis-Tris gradient gels (Invitrogen) according to the manufacturer's instructions. For Western blot, proteins were transferred onto a nitrocellulose membrane, which was subsequently blocked by 3% skimmed milk. All employed antibodies and their applied dilutions are summarized in Table S2.

### Cell culture and immunofluorescence

Immortalized human fibroblast cell lines 1306 (dermal fibroblasts, female Puerto Rican donor) and BJ1-hTERT (foreskin fibroblasts, donor unknown) were kind gifts from Sebastian Iben, Dept. of Dermatology, Ulm University Hospital, Germany. Cells were routinely cultured at 37°C and 5% $CO_2$ in a humidified incubator in DMEM supplemented with 10% FBS (both from Gibco). Antibiotics were only added for selection or maintenance of stable cell lines.

For transfection, the cells were seeded the day before to reach a confluency of 70–80% on the day of transfection and the medium was renewed 30 min before transfection. Transfection of plasmid DNA was performed with Lipofectamine 3000 (Invitrogen/Life Technologies) according to the manufacturer's instructions

except that cells were transfected for 6 h before fresh medium was added. Typically, the cells were allowed to recover and express proteins for 48 h. Stable cell lines were selected by antibiotic treatment for 2 wk with 1 $\mu$g/ml puromycin (Formedium) or 750 $\mu$g/ml geneticin G418 (Formedium). The antibiotic concentration was then lowered to 0.25 or 250 $\mu$g/ml, respectively, and the cells were further cultivated under these conditions. For a mix-clone cell line, fibroblasts were further expanded, and the modification was verified via Western blotting or immunofluorescence. To generate a monoclonal cell line, serial dilution was applied to singulate antibiotically selected cells in 96-well plates. The regrowth of individual clones was monitored until the appearance of larger colonies, and the dilution procedure was repeated if necessary.

Duplex siRNA was transfected using Lipofectamine RNAiMAX (Invitrogen) according to the manufacturer's instructions using 30–40 pmol siRNA per well of a six-well cell culture plate. For protein knockdown, the Lipofectamine–siRNA mix was incubated for 6 h on the cells before medium change and the cells were then incubated for an additional 48 h before experimental use. For cotransfection of siRNA and plasmid DNA, the RNA was already removed 4 h post-transfection. The cells were then allowed to regenerate for 2 h before a second transection of DNA with Lipofectamine 3000 overnight. The growth medium was renewed the day after, and cells were incubated for a further 36–48 h.

For live-cell microscopy, cells were seeded at least 24 h before microscopy in an eight-well culture chamber with a coverslip bottom (Sarstedt). 4–12 h before microscopy, the growth medium was removed and replaced by FluoroBrite DMEM (Invitrogen) supplemented with 10% (wt/vol) FBS and 1% (vol/vol) GlutaMAX (Invitrogen). If applicable, dyes to stain DNA or actin were added according to the manufacturer's recommendations. SPY-DNA (Spirochrome) was used at a dilution of 1:4,000 and SiR-actin

(Spirochrome) at 1:5,000–1:10,000 dilutions with an incubation time of 15 h before microscopy.

Immunofluorescence was performed either on sterile coverslips placed in six-well plates or in eight-well cell culture chambers with cover slide bottom (Sarstedt). The day before, $3.0 \times 10^5$ or $2.5 \times 10^4$ cells were seeded per six-well or eight-well plate, respectively. Cells were washed once with prewarmed PBS immediately followed by covering the surface with 4% (wt/vol) PFA for 10 min. Fixed cells were washed with PBS and permeabilized with 0.1% Triton X-100 in PBS for 20 min. Blocking was performed with 5% (wt/vol) BSA in PBS for 45 min. Primary antibodies were diluted in an antibody diluent (Dako), and a droplet of 25 or 50 $\mu$l was applied on the sample for an eight well or a coverslip upside down, respectively. The staining was performed for 1–4 h at RT or overnight at 4°C in a humidified environment. Costaining with multiple antibodies was performed in a single step, by mixing up to three antibodies in an antibody diluent. Unbound antibodies were removed by washing with PBS before secondary antibodies were applied accordingly for 1 h at RT in a dark environment (typical dilution 1:500 in an antibody diluent). A list of all employed antibodies is provided in Table S2. After washing, cells were coated with mounting medium (Dako), covered with a coverslip (eight-well slide chamber) or a glass slide (coverslips), and stored at least for 6 h at 4°C in the dark until microscopy.

### Generation of a SEPT9 CRISPR/Cas9 knockout cell line

We used the error-prone nature of NHEJ (non-homologous end joining) leading to frameshifts in the coding sequence to generate a SEPT9 knockout cell line in 1306 fibroblasts following an already established protocol (Ran et al, 2013) with some modifications. To identify specific and efficient target sites, the CRISPR design tool in Benchling (https://www.benchling.com/) was used for identification and optimization of small guide RNAs (sgRNAs/gRNAs). The predicted efficiency was based on a scoring system that calculated the on-target and off-target value of each gRNA (Doench et al, 2016). An off-target value above 65, and an on-target above 60 were considered as good guides. To enhance the cleaving efficiency, single gRNAs were individually guiding Cas9 to the beginning of exon 4B and to the end of exon 6E of SEPT9, respectively (template sequence: ENSG00000184640) (Fig S3A). For cloning into the pSpCas9(BB)2A-GFP plasmid (Ran et al, 2013), BbsI overhangs were added to the respective sequences. The resulting sgRNA sequences are shown in Table S3.

We replaced the ORF of the GFP marker in the pSpCas9(BB)2A-GFP plasmid (plasmid #48138; Addgene) by the ORF of iRFP670, resulting in pSpCas9(BB)2A-iRFP670. The forward and reverse sgRNA oligonucleotides were annealed, and the resulting sgRNA insert targeting exon 4B was cloned into *Bbs1*-restricted pSpCas9(BB)2A-GFP, and the sgRNA insert targeting exon 6E was cloned into *Bbs1*-restricted pSpCas9(BB)2A-iRFP670. Annealing and cloning were performed as described elsewhere (Ran et al, 2013).

The plasmids were transfected into 1306 fibroblasts and sorted by FACS 48 h post-transfection. GFP- and iRFP670-positive cells were singulated and sorted into 96-well plates previously seeded with mitomycin C–treated MEF feeder cells (kindly provided by Prof. M. Füchtbauer, Aarhus, Denmark). Upon colony formation, the individual clones were expanded and the DNA was isolated using QuickExtract DNA Extraction Solution (Lucigen). A successful homozygous genomic modification was confirmed by PCR (data not shown) and the absence of respective translation product by Western blotting using antibodies against all long SEPT9 isoforms (Fig S3B and C) and against the G domain only (Fig S3D–F). We used herein two selected clones with identical properties (clone C1 [2c13] and clone C2 [6c29]). The expression of GFP-SEPT9 (but not GFP-SEPT2) in these cells could partly restore the SEPT9 filament network (Fig S5D).

### Transwell migration assay (Boyden chamber assay)

Non-confluent cells were detached from a culture dish and counted. 50,000 cells were resuspended in 100 $\mu$l prewarmed DMEM without FBS and transferred into an 8-$\mu$M PET tissue culture plate insert (BrandTech). The insert was placed into a well of a 24-well cell culture plate containing 700 $\mu$l prewarmed DMEM supplemented with 10% FBS. The cells were allowed to attach and migrate for 20–24 h at 37°C. The next day, the insert was washed twice with PBS inside and outside before fixation with 4% (wt/vol) PFA in PBS for 20 min. After two additional washing steps with PBS, the cell nuclei were stained with a 1:2,000 dilution of SPY-DNA dye (Spirochrome) in an antibody diluent (Dako). SPY505-DNA, SPY555-DNA, or SPY650-DNA was used depending on the fusion protein in the respective cell line.

The documentation of migrated and non-migrated cells was performed in two consecutive steps by fluorescence microscopy. First, the transwell inserts were transferred to a glass slide and at least 10 images at random positions were taken. Non-migrated cells on the top layer of the insert membrane were then removed by a cotton swab, and 10 additional images were taken. The function "Find Maxima" in FIJI was used to automatically identify nuclei in an image. The threshold to exclude background signals was set to a prominence >5,000–20,000 ensuring that only one data point per nucleus was generated. The ratio of migrated cells to the total cell count was used to calculate the migratory behavior of the respective cell line.

### Surface reattachment and cell spreading assays

Cell reattachment and spreading were evaluated simultaneously from the same dataset. Non-confluent cells were detached with trypsin, counted, and seeded to a density of $2.0–3.0 \times 10^5$ cells in a six-well plate. Immediately, a photo (t = 0 min) of a random position was taken through the 10x objective on a routine light microscope (Zeiss Axiovert 40C) using a OnePlus 8T camera at 12 MP (1.6 $\mu$m/px) and the plate was placed back in a $CO_2$ incubator at 37°C. The cell morphology was documented accordingly in triplicate every 15–30 min for a total of 450 min. To evaluate the adhesion of cells, the culture plate was gently rocked back and forth under the microscope.

The level of adhesion was classified into five steps from "none" to "complete" (0%, 25%, 50%, 75%, 100%), defined by the fraction of cells adhering to the plastic surface. The progress of spreading was evaluated based on the photos taken at every timepoint. To evaluate the degree of cell spreading, the morphology of cells

incubated for at least 24–48 h was taken as a reference of complete spreading for each evaluated cell line. Again, the level was categorized in five steps from "completely rounded" to "spreading completed" (0%, 25%, 50%, 75%, 100%). The individual values of each triplicate at any timepoint were then used to generate a heatmap based on this five-step categorization.

### Microscopy and image analysis

Fluorescence microscopy was conducted on a Cell Observer Z1 SD confocal microscope (Zeiss) equipped with 488-nm, 561-nm, and 635-nm diode lasers, Plan-Neofluar 10x/0,3, Plan-Neofluar 40x/0,6, and Plan-Apochromat oil 63x/1,4 objectives (Zeiss), a PS1 incubation system (including incubator, heating insert, temp module S, and $CO_2$ module S, all from Pecon), and an Evolve 512 EMCCD camera (Photometrics). Image processing was performed with Zen blue 2.6 (Zeiss) and FIJI 2.1.0.

Live-cell microscopy was conducted using 10x or 40x air objectives with the laser intensity as low as possible to prevent bleaching. Usually, images were taken every 7.5 min with Z-stacks containing 6–10 layers at multiple positions for a total of 24–50 h or as indicated in the video captions. Analysis was either performed manually in FIJI or semi-automated using the tools StarDist, TrackMate, and MotilityLab as described below.

Total cell sizes were determined from images taken at 10x objective magnification on a routine light microscope (Zeiss Axiovert 40C) using a OnePlus 8T camera at 12 MP (1.6 $\mu$m/px). The image scale was measured using a micrometer calibration slide with defined length units. The cell size was determined using the freehand selection tool in FIJI along the cortex to measure the total area.

For analysis of filopodia, randomly selected z-stack maximum intensity projection fluorescence microscopy images of the bottom from VASP (Invitrogen/Thermo Fisher Scientific)- and/or phalloidin (Thermo Fisher Scientific)-stained cells were analyzed. Filopodium lengths were determined manually in ImageJ 1.53f using the "Line" and "Measure" tools from the filopodium basis to its respective tip.

The size of individual focal adhesions was determined from fluorescence microscopy images using paxillin (SCBT) as a focal adhesion marker in migrating cells. Average intensity z-stack projections of 6–10 images were generated and manually analyzed in ImageJ 1.53f using the "Line" and "Measure" tools.

### Processing of live microscopy data and single-cell tracking

The analysis of time-lapse microscopy data for automated identification of nuclei and tracking of cell movement over time was performed by combining the analysis tools StarDist (Schmidt et al, 2018) (tool for segmentation of cell nuclei in 2D images or stacks based on artificial intelligence), TrackMate (Tinevez et al, 2017) (FIJI plugin to perform single particle tracking of spot-like structures over time), and MotilityLab (http://www.motilitylab.net/) (quantitative and statistical tool to perform cell track analyses).

To prepare microscopy data for the semi-automated process, in each recorded file the nuclear fluorescence channel was extracted, a "Max Intensity" Z-projection was performed in FIJI, and the file was separately saved in .tif format. If required for proper target recognition by StarDist, additional contrast enhancement and background subtraction were performed in FIJI.

StarDist is an open-source package for training and implementation of artificial intelligence (AI) approaches to microscopy imaging. First, microscopic imaging of nuclei was performed. We then trained the AI with either available training datasets (Jukkala & Jacquemet, 2020) or training datasets with preset parameters determining the number of iterations, and the parameters to improve the recognition of nuclei, as well as the progress and validation during the training. Parameters were set as follows: number_of_epochs 400, number_of_steps 12, percentage_validation 10, grid_parameter 2, batch_size 4, patch_size 1,024, n_rays 32, initial_learning_rate 0.0003.

In the next step, the AI was evaluated and tested on validity and generalizability, again using the provided datasets (see link above). A network was described as completely trained once the given curves for "Training loss" and "Validation loss" flattened out. The verified model could then be used to generate predictions from own images. For time-lapse microscopy, the data type was set to "Stacks" representing one image per timepoint. To allow further processing of generated data in TrackMate, the outputs "Region_of_interest," "Mask_images," and "Tracking-file" were generated. The final dataset could then be downloaded as .tif-files and used in TrackMate for further processing. The dataset generated by StarDist consists, among others, of cell tracking files. Each image contains information about the centers of all detected nuclei within a single timepoint represented as a spot. These spots are recognized by TrackMate and tracked over time. Information on pixel or temporal dimensions is not transferred to tracking files by StarDist and was thus manually entered into the image properties in FIJI. Typical values of data generated by our microscope were a pixel width/height of 1.3333 micron, a voxel depth of 5.00, and a frame interval of 450.00 s. The tracking file could then be imported to TrackMate with the correct calibration settings. The tracking of individual nuclei was based on the LoG (Laplacian of Gaussian) detector to identify maxima in the images with too close maxima being suppressed. As only the center of individual nuclei was represented in the file, an estimated blob diameter of 1.00 micron and a threshold of 1.00 with subpixel localization could be used for cell detection. The tracking of nuclei was based on the Linear Assignment Problem algorithm (Jaqaman et al, 2008). Briefly, this algorithm first links spots from frame to frame to generate track segments. Events such as splitting, merging, or gap closing in case of missing detections are then analyzed in a second step. Typical values for frame linking, gap closing, and segment splitting or merging were 15.00 micron with a maximum frame gap of 2. The result contained the individual tracks of each cell nucleus over time allowing to save the data as .xml file for further statistical analysis in MotilityLab or to generate plots using Chemotaxis and Migration Tool (Ibidi). The online tool MotilityLab (freely available at http://2ptrack.net/plots.php) was used to inspect tracks generated by TrackMate and to exclude possible outliers. Data files from batch analyses can simply be imported and quantified online. We used this platform to organize, rearrange, and extract subinformation from the complete dataset and to arrange data for plotting of the MSD and mean track speed. Further statistical analysis was performed manually using Prism (GraphPad Software).

## Statistical analysis

All statistical analyses were based on datasets of at least three independent experiments (triplicate), if not stated otherwise. Individual data points are shown in each graph, and where applicable, the number of measurements or single values is given in the figure caption. Generally, the mean is presented in each graph and error bars represent the SD and are shown whenever possible. Details on statistical analyses including parametric/nonparametric, post hoc test and the multiplicity-adjusted $P$-value are provided in the respective figure and figure caption. Each dataset was tested for potential outliers using the ROUT (robust regression and outlier removal) method. Briefly, this algorithm is based on the false discovery rate. A predicted model fit is then used to decide whether a data point is far enough away to be considered an outlier. With this method, multiple outliers can be identified and removed simultaneously.

If applicable, datasets were tested for normality and lognormality before statistical analysis. Typically, the D'Agostino–Pearson normality test was applied, which first calculates the skewness and kurtosis relative to a Gaussian distribution.

Subsequently, a $P$-value is calculated from the discrepancies of expected and measured values. For $P > 0.05$, a dataset was defined to follow a normal distribution. If no test for normal distribution was performed and datasets were assumed not to follow a Gaussian distribution, a non-parametric test was conducted; otherwise, either a $t$ test or ANOVA was performed.

For distributions that approximately followed a Gaussian distribution, a $t$ test (two groups) or an ANOVA (three or more sets of measurement) was applied. As most datasets were obtained from at least three different groups, commonly a one-way ANOVA was performed, followed by an appropriate post hoc test. Based on these multiple comparison test, a multiplicity-adjusted $P$-value was calculated for each pair. An alpha threshold of 0.05 and a confidence level of 95% have usually been chosen, if not stated differently. Where possible, multiplicity-adjusted $P$-values are shown as number in the figure or figure caption. In the interest of legibility, $P$-values were partially represented as asterisks and described in the corresponding figure caption ($P$-value style GP: 0.1234 [ns], 0.0332 [*], 0.0021 [**], 0.0002 [***], <0.0001 [****]).

The required sample size n for each analysis was calculated a priori with given a error = 0.05, Power = 1-b = 0.98 (calculated with G*Power [v.3.1.9.7]), and a previously calculated effect size f based on 10 randomly selected data points to estimate means and SD for each dataset.

# Supplementary Information

# Acknowledgements

Wieland Mayer and Matteo Hofmann are acknowledged for technical assistance. The authors sincerely thank Peter Krauss for the donation of the anti-SEPT9 antibody, Ernst-Martin and Annette Füchtbauer for providing the feeder cells, and Elias Spiliotis for the EGFP-SEPT9 overexpression plasmid. M Hecht received financial support from the International Graduate School in Molecular Medicine, Ulm University.

## Author Contributions

M Hecht: conceptualization, data curation, formal analysis, validation, investigation, visualization, methodology, and writing—review and editing.
N Alber: data curation and methodology.
P Marhoffer: data curation and methodology.
N Johnsson: conceptualization, funding acquisition, validation, and writing—review and editing.
T Gronemeyer: conceptualization, formal analysis, supervision, validation, investigation, project administration, and writing—original draft.

## Conflict of Interest Statement

The authors declare that they have no conflict of interest.

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
