## [Reviewer comments · Life Science Alliance]

Life Science Alliance

The concerted action of SEPT9 and EPLIN modulates the adhesion and migration of human fibroblasts

Matthias Hecht, Nane Alber, Pia Marhoffer, Nils Johnsson, and Thomas Gronemeyer

DOI: <https://doi.org/10.26508/lsa.202201686>

Corresponding author(s): Thomas Gronemeyer, University of Ulm

Review Timeline:

Submission Date:	2022-08-23
Editorial Decision:	2022-09-30
Revision Received:	2023-11-07
Editorial Decision:	2023-12-01
Revision Received:	2024-04-23
Editorial Decision:	2024-04-25
Revision Received:	2024-04-29
Accepted:	2024-04-29

Transaction Report:

September 30, 2022

Re: Life Science Alliance manuscript #LSA-2022-01686-T

Dr. Thomas Gronemeyer
Ulm University
Molecular Genetics and Cell Biology
James Franck Ring N27
Ulm, Germany 89081
Germany

Dear Dr. Gronemeyer,

Thank you for submitting your manuscript entitled "The concerted action of SEPT9 and EPLIN modulates the adhesion and migration of human fibroblasts". The manuscript has been evaluated by expert reviewers, whose reports are appended below. Unfortunately, after an assessment of the reviewer feedback, our editorial decision is against publication in Life Science Alliance.

Although your manuscript is intriguing, I feel that the points raised by the reviewers are more substantial than can be addressed in a typical revision period. If you wish to expedite publication of the current data, it may be best to pursue publication at another journal.

Given the interest in the topic, I would be open to re-submission to Life Science Alliance of a significantly revised and extended manuscript that fully addresses the reviewers' concerns and is subject to further peer review. If you would like to resubmit this work to Life Science Alliance, you may submit an appeal directly through our manuscript submission system.

Regardless of how you choose to proceed, we hope that the comments below will prove constructive as your work progresses.

Thank you for thinking of Life Science Alliance as an appropriate place to publish your work.

Sincerely,

Reviewer #1 (Comments to the Authors (Required)):

Septins are guanine-nucleotide binding proteins that assemble into oligomers, filaments and higher-order structures. Though septins are required for important cellular processes, such as cell division, adhesion and migration, little is known about non-septin binding partners and the molecular mechanisms that mediate or regulate these functions. Furthermore, septins are known to associate with other cytoskeletal elements such as actin filaments and microtubules, but mechanisms driving their translocation to either element have remained rather elusive.

In this respect the present study by Hecht et al. fills an important gap, as it provides novel and exciting insight into how SEPT9 and its newly identified binding partner EPLIN cooperate to modulate cell adhesion and migration. The authors demonstrate that both proteins interact directly, through EPLIN's LIM domain and the SEPT9 G-domain, and colocalize in fibroblasts. The manipulation of SEPT9 levels is shown to affect cell migration: OE of SEPT9 facilitates, whereas deletion of SEPT9 strongly impairs migration, and defects observed upon loss of SEPT9 can be partially rescued by the concomitant OE of EPLIN. Likewise, cell migration is modulated by EPLIN levels: EPLIN depletion greatly facilitates migration, even in absence of SEPT9, but this effect is strongly diminished upon OE of SEPT9, indicating that SEPT9 fine-tunes EPLIN activities during cell migration. To further corroborate the postulated functional interplay between SEPT9 and EPLIN, however, the authors need to perform rescue experiments, and determine the migratory potential of cells upon re-introduction of FL-EPLIN, or of an EPLIN variant lacking the central, SEPT9-binding LIM domain into EPLIN knockdown cells.

Next, the authors provide evidence that cell adhesion and spreading positively correlate with expression levels of SEPT9, and that the delay in both processes observed in absence of SEPT9 can be partially rescued by OE of EPLIN. Interestingly, and contrary to migration assays, the manipulation of EPLIN levels alone has no impact on adhesion/spreading. The authors should provide an explanation for this apparent inconsistency in the discussion.

To gain insight into the functional interplay between EPLIN and SEPT9 the subcellular distribution of EPLIN was assessed in

presence of variable SEPT9 levels. Under control conditions SEPT9 and EPLIN show limited colocalization, sometimes at the tip of the lamellipodium, sometimes in a subcortical area that seems to be defined by SEPT9. OE or KO of SEPT9 is claimed to alter the thickness of the EPLIN layer, but given the variability in EPLIN/SEPT9 distribution in cells per se the nature of this layer remains unclear to me: Are the EPLIN-positive structures shown in Fig. 5D found at tips of the lamellipodium, or in a subcortical region? How did the authors pick cells for analysis? Based on the same set of experiments the authors claim that the "SEPT9 distribution did not change upon EPLIN OE or KO (see below)". Where in the Results section are these data described, to which figure does this comment refer?

Based on the capability of both, SEPT9 and EPLIN, to associate with actin Hecht et al. then analyzed filopodia length under variable expression levels of the two proteins. EPLIN OE increased filopodia length, but the authors provide no explanation for this phenomenon. How does this occur? Interestingly, however, this effect depends on EPLIN's capability to associate with SEPT9, and on the presence of SEPT9, further pointing towards a functional interplay between the two proteins during actin reorganization. To further confirm this, the authors need to investigate if OE of SEPT9 would restrain the EPLIN OE effect. In addition to filopodia size SEPT9 and EPLIN expression levels also affect other actin-based structures, such as stress fibers, arcs and plasma membrane-associated filaments. But the Results section fails to provide any conclusion that could be derived from observations made under the different conditions. Why does SEPT9 increase the thickness and number of stress fibers? Does SEPT9 (upon OE) localize to these fibers? How is the distribution of OE SEPT9 altered upon manipulation of EPLIN levels? What happens to SEPT9 on actin fibers upon OE of EPLIN lacking the LIM domain?

Finally, Hecht et al. provide evidence that SEPT9 and EPLIN cooperate to modulate focal adhesions. The expression level of either protein is shown to positively correlate with focal adhesion size. Of note, depletion of EPLIN, or deletion of SEPT9 decreased adhesion size, although both conditions led to opposite effects in migration experiments (compare Fig. 3). The authors should comment on this discrepancy.

Overall, the study provides solid evidence that SEPT9 and EPLIN physically interact, and cooperate to modulate the migratory and adhesive behavior of fibroblasts. However, a few of the concerns raised above should be addressed to render the manuscript suitable for publication.

Minor points:

- What do boxed areas in Fig. 5D indicate?
- Page 8: "SEPT9 distribution did not change upon EPLIN OE or KO (see below)" needs to be corrected (KD)
- Page 8: "Cells with EPLIN KD cells" needs to be corrected.

Reviewer #2 (Comments to the Authors (Required)):

In this manuscript the authors investigate the interaction between proteins Septin9 and EPLIN and functional consequences of perturbing the interaction by knockout or over-expression of WT or mutant forms of one or both proteins on cell attachment, spreading and motility of a fibroblast cell line. Having shown effects on cell attachment or motility and that over-expression of Eplin can restore motility of Sept9-KO cells, the authors examine the subcellular localisations of the proteins and report localisations in cell protrusions. From further analyses of actin in protrusions and the sizes and locations of paxillin-containing focal adhesions, the authors conclude that the Sept9/Eplin interaction acts to increase formation of focal adhesions at cell edges, leading to reorganisations of actin filament bundles and thereby effects on cell motility.

Whilst the topic is of interest, interpretations relating to filopodia and focal adhesions are not well-supported by the figures and in places the text is opaque.

General issues. 1. A central limitation is that the entire study is made in a single fibroblast line; the study would be strengthened by demonstration of replication of key experimental results in a separate dermal fibroblast strain.

2. Methods used for measurements of filopodia and focal adhesions do not appear to be explained in the Methods, so there is uncertainty at this time on how to interpret these figure panels.

3. The study does not include 'put back' experiments with re-expression of the relevant cDNA in KO or KD cells. This is particularly an issue for the eplin KD cells, because of potential off-target effects of the siRNA.

4. Although distributions of focal adhesions appear altered in the various KO or over-expression cells, the study lacks mechanistic proof that effects on focal adhesions are a key step leading to altered cell attachment and motility.

Specific Comments.

5. Please include a rationale for picking the 1306 cell line for these experiments.

6. The Methods state that "Details on statistical analyses including parametric/non-parametric, post-hoc test and the multiplicity adjusted P value are given in the respective figure caption", however this reviewer could not find this information in the legends and so some of the graphical data could not be fully reviewed. What do the error bars represent in Fig. 2B? In Fig 2D, are the horizontal lines the means or the medians? Since the data ranges for the WT condition and C1 and C2 KO conditions overlap quite extensively, it is not clear how p values of <0.0001 were arrived at and there is concern that an incorrect statistical method may have been used. Are the 89 cells from one experiment or several? The statistical analysis should be based on n of experiments.

7. There are serious issues with Fig 3B and 3C, as the same datapoints for WT and Eplin KO are included in both graphs, yet these are presented in the text as separate sets of experiments.

8. Fig. 3 legend states data are depicted as mean \pm SD, but the SD are not apparent in the figure. The experimental condition of WT septin9/Eplin OE is missing from Fig. 3C.
9. on p8 and Fig 5 "50% reduced thickness of the Eplin layer". Thickness implies depth, but, from the boxed areas in Fig. 5D, the authors may mean the width of the cortical zone of eplin staining? Please clarify the text. The Sept9 OE cells in Fig. 5D appears to be bi-nucleate, this panel should be replaced by an image of a mono-nucleate cell.
10. Related to Fig. 6A, the authors describe filopodia (in the text) solely on the basis of eplin distribution. Co-staining with a well-established filopodial marker (eg, an actin-bundling protein found in filpodia, or VASP that localises at filopodial tips) is needed to identify these edge structures as filopodia. This is also important because, in the expts reported in Fig. 6C and Fig. 7, where actin staining or co-staining is used, the cells edges appear predominantly as lamellipodia with ruffles and short actin protrusions and there appear to be very few filopodia. A filopodium marker co-stain is needed.
11. In Fig. 6 and 8, graph labellings refer to "filopodia size" and 'focal adhesion size". "Size" is not a precise word here: is filopodia length meant, or focal adhesion area?? The reviewer could not find an explanation of the quantification methods for these structures in the Methods.
12. In Fig. 7, it is curious that the wt/wt cell has very few focal adhesions because focal adhesions are a typical feature of adherent fibroblasts. Also, Fig. 6B reported a similar size range of focal adhesions in wt/wt cells as in eplin-KD, Sept9Wt cells but in Fig 7 the numbers and size of the adhesion appear very different in these two conditions.
13. The effects on focal adhesions and the prominence of focal adhesions in Sept9-over-expressing/elpin KD cells is interesting, but the results shown in Fig. 7 do not match well with the migration results in Fig. 3C. In 3C, KO/KD cells and Sept9WT/elpin-KD cells are both highly migratory but in Fig 7 the appearance of actin and focal adhesions in these conditions is very different. This does not make a convincing case that effects on cell migration are mediated by effects on focal adhesions.
14. A model to explain the effects on F-actin and adhesions is presented with reference to effects on actin-branching and tension-sensing, but this is highly speculative and far beyond the scope of the experiments, as effects on Arp2/3 location or cell tensile forces were not addressed in the experiments.
15. Issues of presentation are noted.
- Please state the sex of the cell line if known.
 - "Quirky" phrasing such as "resembling a fried egg" or "paramecium phenotype" should be removed.
 - In Fig 1C, the 27kda marker is misplaced below the blot panel.
 - The title makes a general statement, but only a single cell line has been used so the phrase "in 1306 cells" would be appropriate in the title.
 - In supplementary Fig. 1A, which Sep9 KO line is used - C1 or C2? This should be stated in every legend for figures where one of these lines has been used.
 - There are various places in the results where the wording is not clear and could lead to mis-interpretation. Checking by a native English speaker would be helpful.
- A non-exhaustive list includes:
- P6 "resulted only in a minor increase of the migration rate..." This is ambiguous because although migration is increased, it remains lower than in WT cells.
- P6 "High levels of SEPT9 could, in contrast, promote cell migration.." - also ambiguous because the resulting motility is not very different to that of the WT/WT cells.
- The results presented in Fig 3C are not clearly described in the results text.
- P7: "around the cortex was restricted by SEPT9 towards the cell center.
SEPT9 and EPLIN showed a partial overlap at this "restriction zone" " - the meaning here is not at all clear. Additional labelling of the cells images would help.
- P9 "The actin cytoskeleton in these cells (Sept9 KO) consisted mainly of filamentous fibers" - this is not apparent from the image, the cells appear to have extensive edge ruffles with small paxillin-adhesions and a band of cortical F-actin.
- P9 "but also translocated paxillin-rich structures from the membrane towards the cytoplasm" - the meaning is not clear and all lefthand panels in Fig 8A show paxillin adhesions in the cell body area as well as at cell edges.

Reviewer #3 (Comments to the Authors (Required)):

In this study, the authors attempted to address the question of how SEPT9 interacts with the actin-binding protein EPLIN and how this interaction controls the migration of human fibroblasts. The strengths of the study include the identification of the domains of SEPT9 and EPLIN involved in their interaction. The authors demonstrated clearly that the G domain of SEPT9 is sufficient for interaction with the LIM domain of EPLIN. Another strength is the comparative analysis of the roles of SEPT9 and EPLIN in cell migration using the same cell type and the same cellular assays. The authors manipulated the cellular levels of SEPT9 (WT, KO, and OE) and of EPLIN (WT, KD, and OE) alone or in various combinations and then examined their impact on the rate of cell migration, cell shape, actin structures, and the size of focal adhesion. The phenotypes were described clearly and quantitatively. Finally, contrary to a previous report, authors demonstrated that EPLIN does not control cell migration solely by inhibiting Arp2/3-mediated actin assembly.

The major weakness of the study is the lack of convincing evidence for the role of the SEPT9-EPLIN interaction in cell migration. The roles of SEPT9 (Fuchtbauer et al., 2011. *Biol Chem*; Dolat et al., 2014. *JCB*) and EPLIN (Karakose et al., 2015. *JCS*; Linklater et al., 2021. *JCB*) in the migration of fibroblasts and/or epithelial cells have been analyzed in terms of focal adhesion and actin dynamics separately in previous studies. The unique contribution of this study is to define the interaction between

SEPT9 and EPLIN and determine the specific role played by this interaction in cell migration. The only thing related to this goal was to overexpress the EPLIN lacking the LIM domain (i.e., the SEPT9-interacting domain) in WT cells and then assess its impact on different aspects of cell migration. The authors showed that the cells expressing the mutant version of EPLIN behaved like the EPLIN-knockdown cells. This raises the possibility that the mutant EPLIN might act in a dominant-negative fashion. Thus, the authors need to come up with more rigorous tests to examine the specific role for the SEPT9-EPLIN interaction in cell migration. Minimally, the authors could show how the EPLIN lacking the LIM domain affects different aspects of cell migration, its own localization, as well as its co-localization with SEPT9 in cells where the endogenous EPLIN is knocked down or depleted.

Despite the interesting speculations and thoughtful discussions, the phenotypes caused by SEPT9 (OE and KO) and EPLIN (OE and KD) appear to be quite distinct and difficult to interpret in terms of SEPT9-EPLIN interaction at the cell protrusions.

Minor points:

- 1) The authors need to describe clearly in the Introduction about what's known on the roles of SEPT9 and EPLIN in cell migration prior to this study.
- 2) While it's mentioned in the text, it is worth emphasizing to the readers that this study is about defining the interaction between SEPT9 and EPLIN and determining its role in cell migration.
- 3) Fig. 1D, it would be informative to include an additional control, i.e., SEPT9 lacking the G domain, in the interaction assay.
- 4) Since EPLIN is not essential for cell migration, the authors could generate EPLIN-KO cells for their study as they did for SEPT9.
- 5) The co-localization between SEPT9 and EPLIN at the cleavage furrow is clear but not so clear at the cell protrusions. In addition, this study involved overexpression of SEPT9-GFP. It seems to be more informative if the authors address the co-localization issue by performing immunofluorescence using antibodies against the endogenous proteins.

Point by point answer Reviewer.

Reviewer 1:

Interestingly, and contrary to migration assays, the manipulation of EPLIN levels alone has no impact on adhesion/spreading. The authors should provide an explanation for this apparent inconsistency in the discussion.

The discussion was amended according to the reviewer's suggestions.

Of note, depletion of EPLIN, or deletion of SEPT9 decreased adhesion size, although both conditions led to opposite effects in migration experiments (compare Fig. 3). The authors should comment on this discrepancy.

We are not fully aware of the issue the reviewer is rising. KO of SEPT9 decreased cell migration approx. to zero. KD of EPLIN could revert this effect. These results were in our opinion adequately discussed.

Are the EPLIN-positive structures shown in Fig. 5D found at tips of the lamellipodium, or in a subcortical region? How did the authors pick cells for analysis?

Cells with clearly visible lamellipodia were selected randomly; no subpopulation was considered. The EPLIN-positive structures are part of the lamellipodium; but not particularly at the tip.

The text remains currently unchanged.

Based on the same set of experiments the authors claim that the "SEPT9 distribution did not change upon EPLIN OE or KO (see below)". Where in the Results section are these data described, to which figure does this comment refer?

This sentence referred to later presented results from Fig. 7. We deleted the respective phrase from this position in the text.

Based on the capability of both, SEPT9 and EPLIN, to associate with actin Hecht et al. then analyzed filopodia length under variable expression levels of the two proteins. EPLIN OE increased filopodia length, but the authors provide no explanation for this phenomenon. How does this occur?

We address this issue now in the discussion.

Interestingly, however, this effect depends on EPLIN's capability to associate with SEPT9, and on the presence of SEPT9, further pointing towards a functional interplay between the two proteins during actin reorganization. To further confirm this, the authors need to investigate if OE of SEPT9 would restrain the EPLIN OE effect.

EPLIN OE in SEPT9 OE cells was investigated and the results were already presented in Fig. 7 and Fig. 8.

We made no changes to the manuscript in this respect.

Why does SEPT9 increase the thickness and number of stress fibers?

The introduction was amended in this respect (direct crosslinking).

Does SEPT9 (upon OE) localize to these fibers?

The localization of SEPT9 to stress fibers was already shown in the literature (e.g. Dolat et al. 2014).

Colocalization of overexpressed SEPT9 with actin stress fibers is shown in Suppl. Fig.S1 and is mentioned in the text.

How is the distribution of OE SEPT9 altered upon manipulation of EPLIN levels?

We added respective IF images of SEPT9 in dependence on EPLIN levels in Fig. S1C.

What happens to SEPT9 on actin fibers upon OE of EPLIN lacking the LIM domain?

We performed additionally IF to address the localization of SEPT9 in cells expressing EPLIN Δ Lim; the results are outlined in the text (see also comments to Reviewer 3).

Reviewer 2:

1. A central limitation is that the entire study is made in a single fibroblast line; the study would be strengthened by demonstration of replication of key experimental results in a separate dermal fibroblast strain.

We repeated some experiments such as the localization of SEPT9 and EPLIN, the migration assay and the quantification of filopodia length in another fibroblast cell line. The results are mentioned accordingly in the text and presented in supplementary figures.

2. Methods used for measurements of filopodia and focal adhesions do not appear to be explained in the Methods, so there is uncertainty at this time on how to interpret these figure panels.

The materials and methods section was amended in this respect.

3. The study does not include 'put back" experiments with re-expression of the relevant cDNA in KO or KD cells. This is particularly an issue for the eplin KD cells, because of potential off-target effects of the siRNA.

We attempted to re-introduce EPLIN Δ Lim in cells expressing EPLIN siRNA, but these cells repeatedly died after transfection. We refrained to continue these rescue experiments. SEPT9 expression in SEPT9 KO cells was already performed; we included this result now in Suppl. Fig. S5 and mentioned it the materials and methods section.

5. Please include a rationale for picking the 1306 cell line for these experiments.

A rationale for the used cell lines is now given in the beginning of the results section.

6. The Methods state that "Details on statistical analyses including parametric/non-parametric, post-hoc test and the multiplicity adjusted P value are given in the respective figure caption", however this reviewer could not find this information in the legends and so some of the graphical data could not be fully reviewed. What do the error bars represent in Fig. 2B? In Fig 2D, are the horizontal lines the means or the medians? Since the data ranges for the WT condition and C1 and C2 KO conditions overlap quite extensively, it is not clear how p values of <0.0001 were arrived at and there is concern that an incorrect statistical

method may have been used. Are the 89 cells from one experiment or several? The statistical analysis should be based on n of experiments.

The statistical analysis description in the methods sections was adapted to appropriately describe the applied procedure. The figure caption of Fig. 2 was extended, addressing the reviewer's concerns. Accordingly, the graph type was changed from a complex-to-interpret violin plot to a scatter plot including means and 95 % confidence interval allowing to visually conclude whether the difference between the means are statistically significant.

7. There are serious issues with Fig 3B and 3C, as the same datapoints for WT and Eplin KO are included in both graphs, yet these are presented in the text as separate sets of experiments.

Data in Fig. 3 A-C are not from separate sets of experiments but from one same set with n=3. The data are presented in separate plots for improved clarity of presentation. The figure caption and manuscript text were amended to clarify this point. Selected reference points from Fig. 3 A & B were also included in Fig. 3 C to allow a clear presentation of statistical relationships between the whole set of protein level modifications. For this reason, equal datasets were equally colored in the graphs. The figure caption and manuscript text were amended to clarify this point. The division in panels A, B, and C was removed.

8. Fig. 3 legend states data are depicted, but the SD are not apparent in the figure. The experimental condition of WT septin9/Eplin OE is missing from Fig. 3C.

The mean +/- SD bars are presented now more pronounced.

Since all data are from the same set of experiments (see above), not all conditions from A and B are shown in C. WT SEPT9 vs. EPLIN OE is shown in B.

9. on p8 and Fig 5 "50% reduced thickness of the Eplin layer". Thickness implies depth, but, from the boxed areas in Fig. 5D, the authors may mean the width of the cortical zone of eplin staining? Please clarify the text. The Sept9 OE cells in Fig. 5D appears to be bi-nucleate, this panel should be replaced by an image of a mono-nucleate cell.

The text and figure caption were changed according to the reviewer's suggestion.

The cells in Fig. 5D show multiple cells in one image, thus multiple nuclei are visible. To clarify this, red dotted lines were drawn around each cell of interest.

10. Related to Fig. 6A, the authors describe filopodia (in the text) solely on the basis of eplin distribution. Co-staining with a well-established filopodial marker (eg, an actin-bundling protein found in filpodia, or VASP that localises at filopodial tips) is needed to identify these edge structures as filopodia.

We performed filopodia staining with anti-VASP, confirming we picked the correct structures in 1306 cells for analysis. However, we refrained to record a full new dataset of all used cell lines for analysis. In BJH cells, we performed the filopodia quantification using the anti-VASP immunostaining, confirming the results from the 1306 cells.

All results are summarized in Suppl. Fig. S5 and are included in the text where appropriate.

11. In Fig. 6 and 8, graph labellings refer to "filopodia size" and 'focal adhesion size". "Size" is not a precise word here: is filopodia length meant, or focal adhesion area?? The reviewer could not find an explanation of the quantification methods for these structures in the Methods.

See comment to point 2.

12. The effects on focal adhesions and the prominence of focal adhesions in Sept9-over-expressing/eplin KD cells is interesting, but the results shown in Fig. 7 do not match well with the migration results in Fig. 3C. In 3C, KO/KD cells and Sept9WT/eplin-KD cells are both highly migratory but in Fig 7 the appearance of actin and focal adhesions in these conditions is very different. This does not make a convincing case that effects on cell migration are mediated by effects on focal adhesions.

See comments to Reviewer 1.

14. A model to explain the effects on F-actin and adhesions is presented with reference to effects on actin-branching and tension-sensing, but this is highly speculative and far beyond the scope of the experiments, as effects on Arp2/3 location or cell tensile forces were not addressed in the experiments.

The model is indeed only a suggestion and is addressed as such in in the discussion. We consider it as a valuable addition since it opens space to discussion and further investigation by us and others.

Since the other two reviewers did not comment on this issue, we would like to keep the model including figure.

15. Issues of presentation are noted.

a. Please state the sex of the cell line if known.

More elaborate details regarding the used cell lines are provided now in the Materials section.

b. "Quirky" phrasing such as "resembling a fried egg" or "paramecium phenotype" should be removed.

The respective phrases were removed.

c. In Fig 1C, the 27kda marker is misplaced below the blot panel.

The figure was corrected.

d. The title makes a general statement, but only a single cell line has been used so the phrase "in 1306 cells" would be appropriate in the title.

We disagree here with the reviewer. Adding the identity of the cell line in the title would make the title less comprehensive. Furthermore, we included now results from another cell line. We would like to leave the title unchanged.

e. In supplementary Fig. 1A, which Sep9 KO line is used - C1 or C2? This should be stated in every legend for figures where one of these lines has been used.

Since different clones of the SEPT9 KO cell line showed insignificant differences in cell mobility (Fig.2), all subsequent experiments were performed only with clone C1.

This statement is now made in the manuscript text.

Clone C2 (and other clones generated but not mentioned here) was kept as backup.

Reviewer 3

Minimally, the authors could show how the EPLIN lacking the LIM domain affects different aspects of cell migration, its own localization, as well as its co-localization with SEPT9 in cells where the endogenous EPLIN is knocked down or depleted.

We have already presented results regarding the migratory behavior of EPLIN Δ Lim overexpressing cells (see Video 4).

We performed additionally IF to address the localization of SEPT9 in these cells; the results are outlined in the text.

We attempted to re-introduce EPLIN Δ Lim in cells expressing EPLIN siRNA, but these cells repeatedly died after transfection. We refrained to continue these rescue experiments.

Minor points:

1) The authors need to describe clearly in the Introduction about what's known on the roles of SEPT9 and EPLIN in cell migration prior to this study.

The introduction was extended in this respect.

2) While it's mentioned in the text, it is worth emphasizing to the readers that this study is about defining the interaction between SEPT9 and EPLIN and determining its role in cell migration.

The introduction was extended in this respect; we added a phrase to the abstract, too.

3) Fig. 1D, it would be informative to include an additional control, i.e., SEPT9 lacking the G domain, in the interaction assay.

The N- and C-terminal extensions of SEPT9 are highly flexible. The expression of these extensions alone would in our point of view not make a good control. Thus we disagree with the reviewer in this respect and would like to keep Fig. 1D unchanged.

4) Since EPLIN is not essential for cell migration, the authors could generate EPLIN-KO cells for their study as they did for SEPT9.

The generation and validation of a EPLIN CRISPR KO cell line including the repetition of all assays performed in the siRNA cells would take several months. This effort is in our point of view beyond the scope of a revision.

5) The co-localization between SEPT9 and EPLIN at the cleavage furrow is clear but not so clear at the cell protrusions. In addition, this study involved overexpression of SEPT9-GFP. It seems to be more informative if the authors address the co-localization issue by performing immunofluorescence using antibodies against the endogenous proteins.

Staining of the endogenous proteins is now shown in the new Suppl. Fig. S1.

December 1, 2023

Re: Life Science Alliance manuscript #LSA-2022-01686-TR-A

Dr. Thomas Gronemeyer
University of Ulm
Molecular Genetics and Cell Biology
James Franck Ring N27
Ulm, Germany 89081
Germany

Dear Dr. Gronemeyer,

Thank you for submitting your revised manuscript entitled "The concerted action of SEPT9 and EPLIN modulates the adhesion and migration of human fibroblasts" to Life Science Alliance. The manuscript has been seen by the original reviewers whose comments are appended below. While the reviewers continue to be overall positive about the work in terms of its suitability for Life Science Alliance, some important issues remain.

Our general policy is that papers are considered through only one revision cycle; however, we are open to one additional short round of revision. Please note that I will expect to make a final decision without additional reviewer input upon re-submission.

Please submit the final revision within one month, along with a letter that includes a point by point response to the remaining reviewer comments.

To upload the revised version of your manuscript, please log in to your account: <https://lsa.msubmit.net/cgi-bin/main.plex>
You will be guided to complete the submission of your revised manuscript and to fill in all necessary information.

B. MANUSCRIPT ORGANIZATION AND FORMATTING:

Sincerely,

Reviewer #1 (Comments to the Authors (Required)):

The manuscript by Hecht et al. on the functional interplay between SEPT9 and EPLIN during focal adhesion turnover and

migration has been significantly improved.

However, one of the concerns initially raised by this reviewer has not been addressed in the revised manuscript: The authors need to validate the specificity of the EPLIN-targeting siRNA used throughout the study. This could be done by rescue experiments, in which the migratory potential of cells is determined upon re-introduction of FL-EPLIN. Ideally, in the same setting a variant lacking the LIM domain should not be able to rescue. This is of significance, as the deltaLIM mutant still displays some colocalization with SEPT9 (Fig.S5E/G: might reflect that EPLINdeltaLIM retains some affinity for SEPT9, or that both proteins localize to actin independently of each other).

In their rebuttal letter the authors indicate that overexpression or EPLIN lacking the LIM domain is toxic to cells in a knockdown background. Does this also hold true for the FL version? Alternatively, a second siRNA could be used to substantiate the claims that EPLIN KD increases the migratory potential of fibroblasts (Fig.3) and affects the organization of the actin cytoskeleton (Fig.7).

Reviewer #2 (Comments to the Authors (Required)):

The authors have done a thorough job in addressing issues noted in my first review, with regard to improving the quality of the Methods section, clarity of presentation and conducting additional experiments.

One further minor text revision - Fig. 3 legend states "EPLIN KD can restore the migratory potential of SEPT9 KO cells" but the observed result is more than a 'restoration" as the migration is increased above that of the wild-type cells and is at a similar level to that of the EPLIN KD (single KD) cells. This result is explained in the Results text, but needs to be more accurately captioned in the figure legend.

Referee cross-comments.

1. My request for the minor text revision remains unchanged.
2. I tend to agree with reviewer 1 that, for a fully rigorous study, authors would need to either validate the specificity of the EPLIN-targeting siRNA used throughout the study. or corroborate with a second independent siRNA that EPLIN KD increases the migratory potential of fibroblasts (Fig.3)

Reviewer #3 (Comments to the Authors (Required)):

The authors have satisfactorily addressed all my concerns. The revision has significantly improved the manuscript. The quantitative analysis of SEPT9 and EPLIN in terms of dosage-dependent cell adhesion and migration, their mutual control of localization, and actin-based structures is rigorous and informative. This study advances our mechanistic understanding of the interplay between the septin and actin cytoskeleton in cell migration.

April 25, 2024

RE: Life Science Alliance Manuscript #LSA-2022-01686-TRR

Dr. Thomas Gronemeyer
University of Ulm
Molecular Genetics and Cell Biology
James Franck Ring N27
Ulm, Germany 89081
Germany

Dear Dr. Gronemeyer,

Thank you for submitting your revised manuscript entitled "The concerted action of SEPT9 and EPLIN modulates the adhesion and migration of human fibroblasts". We would be happy to publish your paper in Life Science Alliance pending final revisions necessary to meet our formatting guidelines.

- please be sure that the authorship listing and order is correct
- please add the Twitter handle of your host institute/organization as well as your own or/and one of the authors in our system
- please use the [10 author names et al.] format in your references (i.e., limit the author names to the first 10)
- please add your main, supplementary figure, and table legends to the main manuscript text after the references section
- please add callouts for Figures 1E; S3A-F; S4A-B and S7A-B your main manuscript text

Figure Checks:

- please indicate in the legend what the dashed and solid vertical lines through the blots are for in Figure 1C and D. Same for Figure S3E.

A. FINAL FILES:

B. MANUSCRIPT ORGANIZATION AND FORMATTING:

Thank you for your attention to these final processing requirements. Please revise and format the manuscript and upload materials within 4 days.

Sincerely,

April 29, 2024

RE: Life Science Alliance Manuscript #LSA-2022-01686-TRRR

Dr. Thomas Gronemeyer
University of Ulm
Molecular Genetics and Cell Biology
James Franck Ring N27
Ulm, Germany 89081
Germany

Dear Dr. Gronemeyer,

Thank you for submitting your Research Article entitled "The concerted action of SEPT9 and EPLIN modulates the adhesion and migration of human fibroblasts". It is a pleasure to let you know that your manuscript is now accepted for publication in Life Science Alliance. Congratulations on this interesting work.

DISTRIBUTION OF MATERIALS:

Again, congratulations on a very nice paper. I hope you found the review process to be constructive and are pleased with how the manuscript was handled editorially. We look forward to future exciting submissions from your lab.

Sincerely,
